# CHATANI: LANGUAGE-DRIVEN MULTI-ACTOR ANIMATION GENERATION IN STREET SCENES

## ABSTRACT

Generating interactive and realistic traffic participant animations from instructions is essential for autonomous driving simulations. Existing methods, however, fail to comprehensively address the diverse participants and their dynamic interactions in street scenes. In this paper, we present ChatAni, the first system capable of generating interactive, realistic, and controllable multi-actor animations based on language instructions. To produce fine-grained, realistic animations, ChatAni introduces two novel animators: PedAnimator, a unified multi-task animator that generates interaction-aware pedestrian animations under varying task plans, and VehAnimator, a kinematics-based policy that generates physically plausible vehicle animations. For precise control through complex language, ChatAni employs a multi-LLM-agent role-playing approach, using natural language to plan the trajectories and behaviors of different participants. Extensive experiments demonstrate that ChatAni can generate realistic street scenes with interacting vehicles and pedestrians, benefiting tasks like prediction and understanding. All related code, data, and checkpoints will be open-sourced.

## 1 INTRODUCTION

Realistic and diverse street scene animations of traffic participants serve in various fields, including game, movie, especially autonomous driving simulations Caesar et al. (2020); Sun et al. (2020); Xiao et al. (2021); Dosovitskiy et al. (2017). The participants' animation reflect their temporal dynamics and motion. Different behaviors and interactions of traffic participants constitute the temporal events within the scene, which contain crucial temporal information. Perceiving, understanding, and predicting the trajectories and motions of traffic participants Sun et al. (2020) composed of animations is a key step in driving system and making decisions within them. The animation incorporates the participants' motion representation, thereby serving as a prior to drive the rendering engine or diffusion model for the final simulation. Therefore, constructing diverse and high-quality animation is a key component of driving simulators. However, how to generate pedestrian and vehicle animations with interactive attributes remains an unsolved problem.

In driving simulator like Dosovitskiy et al. (2017), animations are manually generated as templates, which is inefficient, not diverse, and especially not realistic. Some existing works adopt learning-based approaches to address physical and dynamic infeasibility or to generate more reasonable behaviors, thereby improving the quality of animations. However, all of these methods overlook the interactivity among participants in street scenes. The generation process and the occurrence of interactions typically exist on two levels: the high-level, which is concerned with trajectory planning, and the low-level, which focuses on detailed

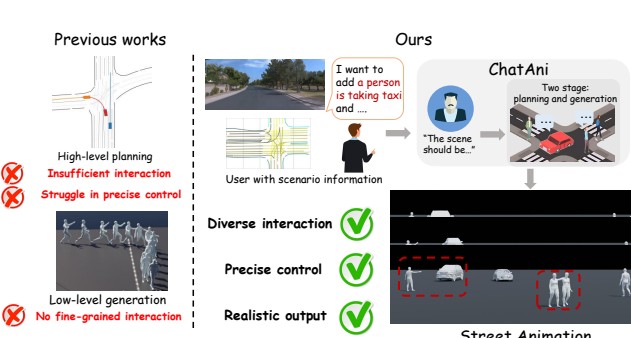

Figure 1: ChatAni generates interactive and realistic multi-actor street animation by language.

high-level, which is concerned with trajectory planning, and the low-level, which focuses on detailed

animations. Existing works have not comprehensively addressed both aspects and the interactive attributes inherent in them. At the high level, LCTGen Tan et al. (2023) and CTG++Zhong et al. (2023a) use language to generate vehicle trajectories, but they do not consider different participants, including pedestrians, and thus lack pedestrian-involved high-level interactive attributes. At the low level, PacerRempe et al. (2023) and Pacer+ Wang et al. (2024) generate fine-grained pedestrian animations but still do not account for potential low-level interactions between multiple pedestrians or between pedestrian and vehicle.

We introduce ChatAni, the first system achieving interactive and realistic language-driven multi-actor street animation generation, as shown in Fig. 1. ChatAni simultaneously introduces interaction designs to both low-level and high-level aspects, ensuring interactivity, realism, and controllability.

In ChatAni, we design specialized low-level animation generators tailored to the distinct characteristics of pedestrian and vehicle motions. For interactive and realistic pedestrian animation, we introduce PedAnimator, a unified multi-task framework that generates physically plausible animations via physics-driven control and diverse input signals. PedAnimator is the first method to achieve physical interactions between pedestrians and pedestrian-vehicle through a novel interaction training strategy that integrates general observations and goals, enabling diverse interactions with minimal adjustments. It supports control over both trajectory and body motion, with policy unification achieved through a task-masking mechanism that enables generation under a single policy. Furthermore, hierarchical control and body-masked adversarial motion priors are incorporated, introducing priors into both action and reward spaces. These not only support multiple control objectives but also enhance the realism and human-likeness of the generated animations.

To generate physically feasible vehicle animations, we introduce VehAnimator, a kinematic-based control policy generating physically feasible vehicle animations. It converts planned raw trajectories into kinematically compliant animations through a dynamic model, eliminating unrealistic artifacts like tail swinging/drifting. The framework ensures interactivity via obstacle-aware training enabling inherent collision avoidance. Temporal-aware designs improve precision and temporal consistency.

To achieve language-driven high-level control and interactive attributes, ChatAni models each traffic participant as an agent and employs a multi-LLM-agent role-playing approach. Agents organize their own information based on requirements and communicate with other agents, ultimately invoking tools to output planning, which are then executed by PedAnimator and VehAnimator. Benefit from the powerful understanding capability and prior knowledge of LLM, the design enables precise control over the entire scene animation generation through natural language, while the communication process between the agents facilitates the planning of high-level interactive attributes.

ChatAni achieves three core features: *interactivity*, *realism*, and *controllability*. For interactivity, low-level animators incorporate interactive attributes, while multi-agent LLM role-playing handles high-level planning. For realism, PedAnimator employs hierarchical control with body-masked adversarial motion priors to generate human-like motions, while VehAnimator simulates real vehicle physics and kinematics. For controllability, LLM agents adhere strictly to language commands, with animators executing actions with precision. The final output animations are suitable for driving simulators, based on either rendering engines or diffusion models.

Our contributions include: (1) the first language-driven system for multi-actor street scene animation generation; (2) PedAnimator enabling interactive pedestrian animations with varied control signals; (3) VehAnimator creating physically plausible animations from trajectories; (4) a multi-LLM-agent framework generating language-controlled interactive high-level plannings; (5) comprehensive experiments validating ChatAni's command-aligned animations with enhanced interactivity/authenticity and benefits to the prediction and understanding tasks.

## 2 RELATED WORKS

**Human animation generation.** Human animation generation can be broadly divided into kinematic and physics-based approaches. For kinematics-based generation, transformer-based methods Athanasiou et al. (2022); Guo et al. (2024; 2022) and diffusion models Zhang et al. (2022; 2023) are used to generate the corresponding animation based on language inputs. Recent methods Shafir et al. (2023); Xie et al. (2023); Wan et al. (2023) introduce further control conditions. However, these approaches do not account for physical constraints. For physics-based generation, existing

work such as Peng et al. (2021; 2022); Won et al. (2022); Luo et al. (2023; 2024) achieves predefined tasks with plausible animations. Tessler et al. (2023); Juravsky et al. (2022); Bae et al. (2023); Xu et al. (2023a;b) extend the animation content with different designs. Rempe et al. (2023); Wang et al. (2024) focus on the animation of pedestrians in the street scene. However, these methods do not involve the interaction behaviors between multi-pedestrians or pedestrian-vehicle. Our PedAnimator considers the interaction behaviors and is trained as a unified policy for multiple scenarios also including following, imitation.

**Vehicle traffic generation.** In industry, vehicle traffic is usually generated by software tools Chen et al. (2022); Queiroz et al. (2019); Fremont et al. (2019); Jesenski et al. (2019). Some research works Bergamini et al. (2021); Tan et al. (2021); Feng et al. (2023); Rempe et al. (2022) focus on unconditioned generation to simplify the generation process. Recent works on vehicle traffic generation Suo et al. (2021); Zhong et al. (2023b;a); Tan et al. (2023); Lu et al. (2024); Ding et al. (2023) introduce different conditions to the process. However, these works generally do not directly consider physical and kinematic constraints for the traffic and lack sufficient interaction. Several early studies, such as those by Li et al. (2017); Lin et al. (2016); Wang et al. (2018), consider certain potential interaction behaviors in traffic. However, these works are unable to achieve effective control through means such as linguistic commands, and they also lack sufficiently granular modeling of interactions involving pedestrians, such as hailing a taxi or vehicle-pedestrian collisions. And these language-controlled methods cannot achieve precise control over different vehicles or other participants. Our LLM-agents planning utilizes the language understanding capability of LLMs to achieve precise control of participants considering interactive information, and VehAnimator generates the final physical plausible vehicle animation.

**Large language models and agents.** Recently, numerous large language models Touvron et al. (2023); Liu et al. (2024); Bai et al. (2023); Achiam et al. (2023) have been proposed and released. Many works have used these models by fine-tuning themHu et al. (2021); Qiu et al. (2023) or integrating them with relevant tools to build LLM-based agents Liu et al. (2023); Wu et al. (2023). These agents have been applied to a wide range of downstream tasks Hong et al. (2023); Zhou et al. (2023); Li et al. (2024b;a); Leng & Yuan (2023); Shen et al. (2024). In this paper, we explore the application of agents in traffic simulation, using them to interact and perform trajectory and behavior planning within traffic scenarios.

## 3 METHOD

ChatAni is the first system to enable interactive, realistic, language-driven multi-actor animation generation for street scenes, composed of three components: (1) PedAnimator—a unified multi-task controller that produces physically plausible pedestrian animations by integrating interaction signals, trajectory tracking, and upper-body motion. It incorporates interaction task training, task-masking with embeddings for policy unification, Body-masked Adversarial Motion Priors, and hierarchical control to enhance animation quality; (2) VehAnimator—a vehicle animation generator that translates high-level raw trajectories into physically feasible vehicle motions via kinematic constraints and action-to-state control; and (3) a Multi-LLM-Agent Role-Playing Framework—a language-driven planner leveraging LLMs' natural language understanding to construct role-specific interactive agents with inter-agent communication, ensuring semantically consistent, language-aligned high-level plans. During execution, traffic participants are initialized as LLM agents that collaboratively establish the scene context through communication. The generated high-level plans are then dispatched to PedAnimator or VehAnimator, which synthesize low-level animations under physical constraints. These outputs are composited into final animations that jointly satisfy physical plausibility and linguistic specifications. The system overview is shown in 2.

### 3.1 PEDANIMATOR

PedAnimator (Fig. 3) is a unified multi-task controller generating interaction-aware pedestrian animations through diverse control signals (interactions, trajectories, upper-body motions). Multi-pedestrian interactions constitute critical event elements in traffic scenarios yet remain unaddressed by existing methods. PedAnimator employs a dedicated interaction training pipeline with universal observation-action frameworks, enabling physical plausibility across interaction tasks via minor objective adjustments. The system also integrates task-masking mechanisms with embeddings for

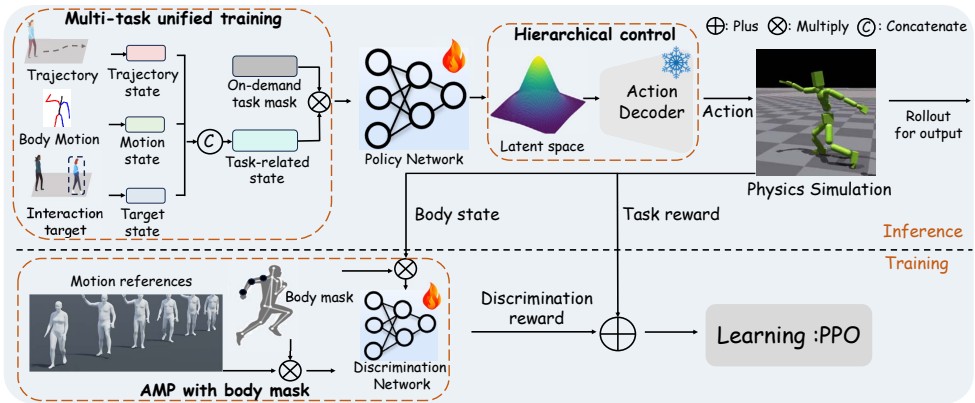

Figure 3: Pedestrian animator (PedAnimator) framework. PedAnimator takes interaction behaviors into account. With multi-task unified training, PedAnimator achieves unified control over various tasks, including following, imitation and interaction. Hierarchical control and AMP with body mask provide prior to action and reward space, improving the realism of animations.

multi-task unification, body-masked Adversarial Motion Priors (AMP), and hierarchical control in reward/action spaces to ensure control fidelity while producing human-like animations that fulfill physical and visual requirements.

The control process is defined by Markov decision process $\mathcal{M}^p = \{\mathcal{S}^p, \mathcal{A}^p, \mathcal{T}^p, \mathcal{R}^p, \gamma^p\}$, where the elements represent states, actions, transition process, reward, and the discount factor. Goal-conditioned reinforcement learning (RL) is adopted for training. The transition process is implemented by the physics engine. Others are detailed below.

**States and Multi-task Unified Training.** Our control policy is designed to handle multiple distinct tasks, requiring specialized processing to achieve a unified policy capable of addressing them. The tasks are categorized into three main groups: trajectory following, single-agent behavior specification, and multi-pedestrian interaction. For single-agent behavior, the LLM generates text descriptions, which are then processed by a Text2Motion model (e.g., Mo-Mask Guo et al. (2024)) to produce upper-body motion for the policy to replicate, specifying single-agent behavior. Multi-agent interaction tasks are trained based on predefined types, with the LLM classifying interaction behaviors, enabling the animator to execute specified interactions.

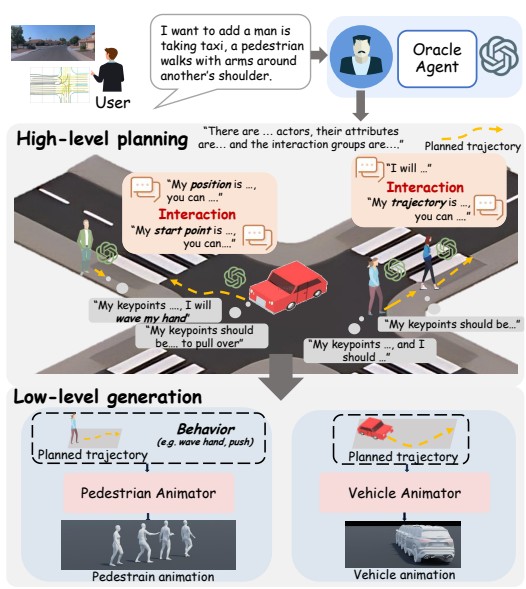

Figure 2: ChatAni adopts multi-LLM-agent role-playing for precise high-level planning. Two specialized animators are designed for low-level animation.

Task-related states consist of trajectory slices $\mathcal{S}^p_{traj}$, motion to be imitated $\mathcal{S}^p_{mo}$, and states of interacting targets $\mathcal{S}^p_{tar}$. $\mathcal{S}^p_{traj}$ includes the K future steps of the planned trajectory, while $\mathcal{S}^p_{mo} = \text{concat}(\hat{j}_{pos}, \hat{j}_{rot}, \hat{j}_{vel}, \hat{j}_\omega)$ represents joint position $\hat{j}_{pos}$, rotation $\hat{j}_{rot}$, velocity $\hat{j}_{vel}$, and angular velocity $\hat{j}_\omega$, with optional joint masking to focus on relevant parts, such as the upper body. $\mathcal{S}^p_{tar} = \text{concat}(r_{pos}, r_{rot}, r_{vel}, r_\omega, r_{bbox}, p_{inter}, e_{inter})$ includes the root position $r_{pos}$, root rotation $r_{rot}$, root velocity $r_{vel}$, angular velocity $r_\omega$, bounding box vertices $r_{bbox}$, interaction contact position $p_{interaction}$, and an interaction embedding $e_{inter}$ as a one-hot vector specifying the interaction characteristic. The states of interacting targets remain consistent across behaviors, with only minor adjustments to control signals, allowing for diverse interactions under a unified framework. The interactive behavior is uniformly modeled. By controlling variables such as the specific interaction points and the intensity of interaction, and by integrating trajectory following and motion imitation,

a diverse range of interactive behaviors can be achieved. In addition to task-related states, the final observation includes humanoid proprioception Wang et al. (2024) $\mathcal{S}^p_{prop}$, capturing the internal states of the humanoid. The final state is represented as $\mathcal{S}^p = \text{concat}(\mathcal{S}^p_{traj}, \mathcal{S}^p_{mo}, \mathcal{S}^p_{tar}, \mathcal{S}^p_{prop})$. To enable specific parts during training or testing, we introduce a task masking mechanism. During training, tasks for each episode are sampled, and the corresponding task-related states are unmasked while others are masked. This prepares only the relevant environment, and rewards are calculated based on these active tasks. During inference, relevant states are activated while others are masked, ensuring task-specific execution. This approach ensures that the unified policy can handle all tasks without confusion. After sufficient training, the policy performs at the level of individually trained policies while being capable of executing multiple non-conflicting tasks in a single run.

**Action hierarchical control.** Pedestrian action spaces are typically modeled using a proportional-derivative (PD) controller at each degree of freedom (DoF), but such spaces lack inherent priors, often leading to locally unrealistic actions for completing specific tasks. To address this, we apply a hierarchical action control space from PULSE Luo et al. (2024). The policy network first outputs to a pretrained latent space, obtaining a latent feature $\mathbf{f_{action}}$, which is then decoded by a pretrained decoder into control signals. The pretrained latent space, equipped with the corresponding decoder, ensures that the decoded output distribution closely matches the input data from pretraining, thereby providing the action space with a real-world action prior.

**Reward design.** The reward consists of two main components. The first is the discrimination reward $R_{disc}$ used to implement AMP Peng et al. (2021), which employs a discriminator to encourage the policy to generate output that aligns with movement patterns observed in a dataset of human-recorded data clips. The second component is the task-related reward $R_{task}$, designed to motivate the policy to accomplish specific tasks and execute high-level planning. The detailed task reward designs can be found in appendix S2.2. Similar to Luo et al. (2023), we implement early termination if an excessive root distance error or joint distance error occurs, and fail-state recovery task for robustness. The recovery task also plays an important role in low-level interactions between pedestrian and vehicle.

**AMP with body mask and warm-up training.** We employ Proximal Policy Optimization (PPO) Schulman et al. (2017) for overall training optimization. However, directly optimizing for interaction tasks presents certain challenges: without AMP, the lack of a discrimination reward makes it difficult to achieve human-like animation results. Conversely, proper reference data clips needed for specific tasks are usually complex and not easily available. Using reference data clips from the following or imitation tasks can lead to misalignment with the actual requirements for interaction, resulting in failures during interaction tasks.

We address these challenges through two strategies: (i) A two-phase approach: initial AMP-free training establishes task-completion foundations for subsequent AMP-enhanced optimization that preserves functionality while improving realism; (ii) AMP body masking that excludes interaction-focused joints (e.g., arms) during discriminator calculations, enabling AMP-driven natural motion for non-interactive body parts while maintaining task-specific joint flexibility. These techniques collectively enable visually realistic animations with effective interaction task performance.

## 3.2 VEHANIMATOR

VehAnimator, as shown in Fig. 4, translates raw vehicle trajectory planning into physically and dynamically feasible animations. Although trajectory data alone could theoretically represent rigid-body vehicle motion, raw trajectories typically lack necessary kinematics constraints, leading to unrealistic phenomena such as tire slippage and drifting when applied directly. As a physics-aware controller, VehAnimator processes trajectory inputs to generate control signals, then applies vehicle animation equations for state

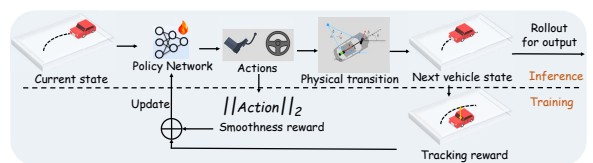

Figure 4: Vehicle animator (VehAnimator) framework. VehAnimator adopts goal-conditioned RL based on physical transition of real vehicle. Combining with history-aware design, VehAnimator generates realistic vehicle animation under planned trajectory.

transitions that produce final animations. This imple-

mentation ensures kinematics validity in the outputs while enhancing control accuracy and temporal consistency through historical-aware action-observation designs. The control process is also modeled as a Markov decision process defined by $\mathcal{M}^v = \{\mathcal{S}^v, \mathcal{A}^v, \mathcal{T}^v, \mathcal{R}^v, \gamma^v\}$, with goal-conditioned RL for training.

**History-aware states.** The VehAnimator incorporates historical information into its states, alongside its current state, to improve the temporal consistency of the policy. The vehicle states consist of the planned trajectory segment $\hat{\mathbf{P}}^v$, temporal velocity $\mathbf{V}^v$, and dynamic parameters $\Theta^v$. $\mathbf{P}^v$ is a slice of the planned trajectory in the vehicle's coordinate system over the current and adjacent $\tau^v$ frames. $\mathbf{V}^v$ represents the vehicle's centroid velocity over $\tau^v$ frames. $\Theta^v$ includes inherent vehicle parameters such as $L$ (vehicle length), $W$ (vehicle width), $l_f$ (front overhang), and $l_r$ (rear overhang). These parameters provide prior information that influences the dynamic transition process.

**Vehicle actions.** To accurately simulate real vehicle actions and maintain temporal consistency, the vehicle action space $\mathcal{A}^v$ is defined by the delta steering angle $\Delta\delta$ and scalar acceleration $a$, which are the two most direct controls affecting vehicle movement in actual driving.

**Vehicle transition function.** We employ the bicycle model to model the vehicle dynamic transition process. Let $x$ and $y$ denote the vehicle's coordinates, $v$ represent the scalar velocity, and $\phi$ indicate the vehicle's orientation. Utilizing the inherent parameters from the states, the vehicle's state transition process can be expressed as:

$$\dot{x} = v\cos(\phi + \beta), \ \dot{y} = v\sin(\phi + \beta), \ \beta = \arctan\left(\frac{l_r}{l_f + l_r}\tan(\delta)\right), \ \dot{\phi} = \frac{v}{l_f + l_r}\cos(\beta)\tan(\delta).$$

Where $\beta$ denotes tire slip angle and $\dot{}$ denotes derivation. The bicycle model effectively simulates the dynamic state changes of the vehicle determined by vehicle actions.

**Reward and training.** The reward focuses on two aspects: following the planned trajectory and smoothness of the results. Therefore, the reward consists of two components: $R_{pos}^v = -||\hat{p}_t - p_t||_2$ for following the planned trajectory and $R_{act}^v = -(c_\delta||\Delta\delta||_2 + c_a||a||_2)$ for smoothness, with $c_\delta$ and $c_a$ are two coefficients to balance the two different units. Training uses TD3 Fujimoto et al. (2018) to maximize the accumulated discounted reward. Note that: (i) actions can be further smoothed with temporal filtering: $\mathcal{A}_t^v = \alpha\mathcal{A}_{t-1}^v + (1 - \alpha)\mathcal{A}_{policy}^v$, where $\mathcal{A}_t$ is the action taken in timestep $t$ and $\mathcal{A}_{policy}$ is the action directly output from policy network; (ii) obstacles can be considered by concatenating their positions and radius in the state and adding $R_{obs}^v = \frac{\epsilon}{L_o}$ to the reward if $L_o < D$, where $L_o$ is the distance to obstacle, $\epsilon$ is a coefficient and $D$ is a threshold.

### 3.3 MULTI-LLM-AGENT ROLE-PLAYING SYSTEM DESIGN

PedAnimator and VehAnimator facilitate precise low-level control for pedestrian and vehicle animations. For language-based holistic scene control, we developed a multi-LLM-agent role-playing framework that generates high-level planning for all participants from user input, offering control signals for low-level animation. The framework includes agents with LLM-based reasoning and tool interfaces, divided into two types: (1) Oracle Agent processes user language inputs to create global scene context and scheduling instructions; (2) Actor Agents represent individual traffic participants, initialized with Oracle Agent information, and engage in inter-agent communication to collaboratively develop high-level plans. This design utilizes the strong language understanding and domain knowledge of LLM to accurately interpret user input. Actor Agents dynamically adjust planning based on participant types (e.g., pedestrians, vehicles), meeting diverse behavioral needs through specific tool functions while ensuring scene-level coordination. See Fig. 2 for details.

#### 3.3.1 ORACLE AGENT

The Oracle Agent decomposes complex user instructions into participant-specific actions, ensuring precise interpretation and schedule generation. Its LLM component is prompted with role definitions, requirements, and few-shot examples, and outputs initialization information, interaction groups, and schedules in natural language. We implement a structured output tool that parses these linguistic outputs into executable predefined data structures. Oracle Agent enables the system to process composite abstract instructions while enhancing operational clarity and granularity. See more details in the appendix S4.

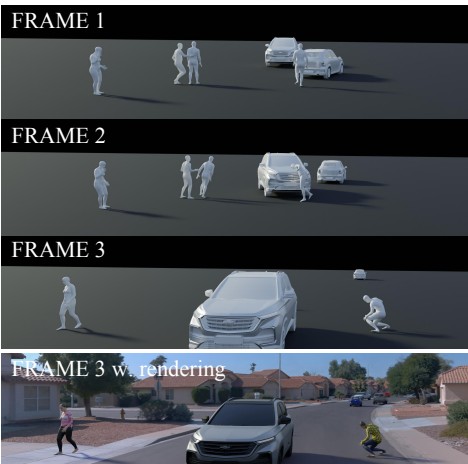

**Command:** *A person is taking a taxi at the left roadside, and a vehicle overtakes the taxi. Two persons are walking together with one's arm around another's shoulder. A person is chasing another person along the roadside.*

**Command:** *A person pushes another person, and another person is making a phone call, then walks along the roadside. A vehicle turns right at the intersection and hits a man, and a hurried vehicle changes lane.*

Figure 5: System results under complex and composite commands, with realistic animation output and diverse interaction information between pedestrian-vehicle, multi-pedestrian, multi-vehicle.

Pedestrian animations of **pushing** task

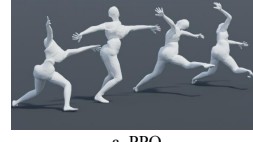 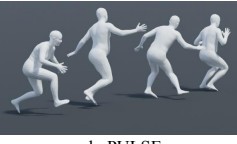 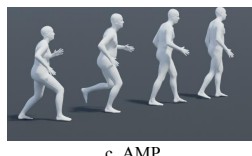 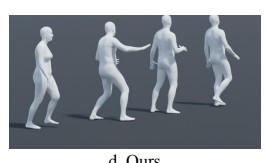

a. PPO        b. PULSE        c. AMP        d. Ours

Figure 6: Visualization comparison of pushing task. PedAnimator complete the task with high visual quality.

### 3.3.2 ACTOR AGENT

Each Actor Agent corresponds to a participant in the scene, initialized with information from Oracle Agent, including agent type, trajectory, and behavioral descriptions. These agents communicate according to the Oracle Agent schedule to collect interactive information and formulate final high-level planning. The planning process incorporates map data, where the scene map is represented as a graph $\mathcal{G} = (\mathcal{N}, \mathcal{E})$. $\mathcal{N}$ denotes lane sections, each containing a point set and metadata such as orientation and driving type (e.g., straight, turn, lane/boundary). Edges $\mathcal{E}$ encode relationships between lane sections (e.g., adjacency, connectivity). The planning adopts a keypoints-based methodology, synthesizing actor interactions to refine the final trajectory.

**Keypoints-based trajectory planning.** Trajectories are defined by keypoints interpolated into continuous paths. The Actor Agent's LLM component determines the required keypoint count and their generation logic based on initialization information. Keypoints originate from two sources: (1) Map retrieval: extracted from the map $\mathcal{G}$ by analyzing attributes (e.g., lane orientation, driving type) and topological relationships (via edges $\mathcal{E}$) between nodes $\mathcal{N}$; (2) Interaction derivation: determined through inter-agent communication, where dependencies on other participants' states influence keypoint placement. This hybrid approach ensures trajectory coherence with both environmental constraints and multi-agent interactions.

**Inter-agent communication.** Communication allows agents to acquire information from others to generate interaction-dependent keypoints. Predefined interaction groups, through the Oracle Agent, guide the process. The LLM identifies the required parameters (e.g., start/end points, trajectories) from interacting agents. Agents exchange requested data, which are processed by tools to generate keypoints. This mechanism supports interactive trajectory generation.

Bézier curve interpolation aggregates keypoints to generate complete planning trajectories, with three operational notes: (1) interpolated trajectories serve as physics-agnostic plans for animators; (2) static actors require two orientation-defining keypoints; (3) pedestrian agents have LLM-inferred behavioral instructions (e.g., "wave hand while walking") from initialization data—derived from explicit commands/implicit semantic context—formatted as executable text directives.

## 4 EXPERIMENTS

### 4.1 IMPLEMENTATION DETAILS AND DATASET

PedAnimator uses Isaac Gym Makoviychuk et al. (2021) as physics simulator, AMASS Mahmood et al. (2019) providing reference data, employing a SMPL Loper et al. (2023) model as the simulated entity. LLM components employ GPT-4 Achiam et al. (2023) API. The final rendering adopts pipeline from ChatSim Wei et al. (2024). PedAnimator and VehAnimator are trained independently in separate environments and models. Detailed settings, configurations, more experiments, video results, and part of codes can be found in the appendix and supplementary materials.

### 4.2 SYSTEM RESULTS

#### 4.2.1 INTERACTIVE, REALISTIC AND CONTROLLABLE

We showcase representative keyframes from two simulated scenarios in Figure 5, each containing various traffic participants generated according to commands. The results demonstrate three critical capabilities: (i) interaction is thoroughly depicted, including pedestrian-vehicle (crashing, taxi hailing, collision avoidance, speed adaptation), vehicle-vehicle (lane switching, overtaking), and pedestrian-pedestrian (pushing, chasing, walking with arm around shoulder), which enrich the scene animations while enhancing the suitability for more diverse and dangerous cases. For instance, pushing is dangerous and corner in the street scenes, which is not easy to collect from real scene and crucial to safety; (ii) realistic low-level animation is achieved by the two physics-based animators, evident in the physically plausible human interaction animation with explicit physical feedback, and robust PedAnimator supports

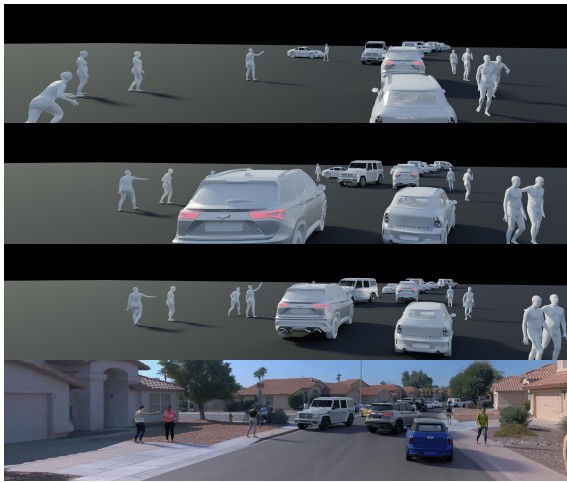

**Command:** *Make the scene crowded like a traffic jam. On the left, one person has slipped, and another is about to help.*

Figure 7: System results under abstract command with large-scale output. ChatAni can handle abstract semantics and scale effectively to manage larger scenarios.

recovery from falling; (iii) precise control is enabled with the multi-LLM-agent framework through decomposition of complex descriptions containing abstract semantics into executable instructions. Combining these features, the generated animations are more realistic and better capture dangerous corner cases that are otherwise difficult to obtain

#### 4.2.2 SCALABLE AND ABSTRACT UNDERSTANDING

We demonstrate ChatAni's ability to understand and decompose abstract semantics in 7, while also supporting more diverse interaction behaviors. At the same time, this result showcases ChatAni's scalability in large-scale scene traffic generation, as it is capable of handling a large number of traffic participants simultaneously present in the scene, constructing crowded and complex traffic scenarios through the interactions of numerous people, vehicles, and their behaviors.

### 4.3 COMPONENT RESULTS

#### 4.3.1 PEDANIMATOR

We evaluate PedAnimator on trajectory following and motion imitation tasks, as shown in Table 1. The quality and diversity of the generated results are measured by Frechet Inception Distance (FID) Heusel et al. (2017) and diversity metric (Div.) Wang et al. (2024) at normal speed, and by l-FID and l-Div. at low speed. Imitation accuracy is measured by Mean Per-Joint Position Error (MPJPE), while following accuracy is evaluated using following error ($E_f$). User preference (UP)

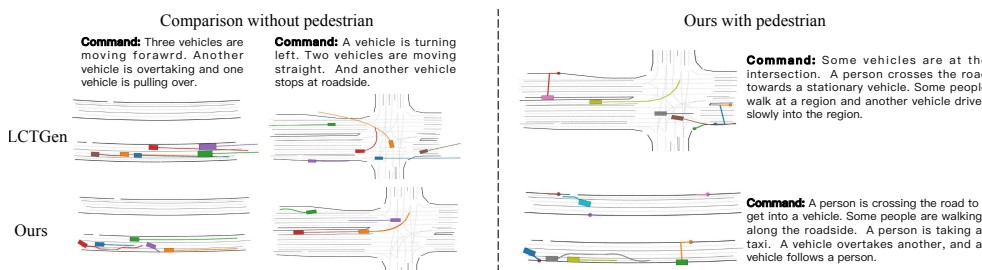

Figure 8: High-level planning results comparison under vehicle-only command, and our planning results with pedestrians involved. The boxes indicate vehicles and the circles indicate pedestrians. Our method significantly and consistently achieves more precise and reasonable plannings.

is assessed by 100 users who evaluate 33 animation segments. Benefit from hierarchical control and its training strategy, PedAnimator generates high-quality animations, demonstrating competitive performance in both following and imitation tasks.

We evaluate interaction task performance across methods, with PedAnimator employing a unified policy versus task-specific training in others (Table 2, Fig. 6). Testing three interaction tasks - pushing, patting, and arm-around-shoulder walking - reveals AMP Peng et al. (2021) produces reference-like motions but lacks task-specific data, yielding superficial mimicry without task completion. While PPO Schulman et al. (2017) and PULSE Luo et al. (2024)

| Methods | FID↓ | Div.↑ | l-FID↓ | l-Div.↑ | $E_{mpjpe}$↓ | $E_f$↓ | UP↑ |
|---|---|---|---|---|---|---|---|
| Pacer | 7.25 | 1.24 | 7.93 | 1.05 | \ | 0.122 | 0.171 |
| Pacer+ | 6.62 | 1.58 | 7.76 | 1.28 | 82.33 | 0.128 | 0.284 |
| Ours | **6.21** | **1.76** | **7.07** | **1.49** | 79.82 | 0.124 | **0.545** |

Table 1: Animation quality and accuracy evaluation for trajectory following and motion imitation.

| Methods | Unified policy | Interaction 1 | Interaction 2 | Interaction 3 |
|---|---|---|---|---|
| PPO | ✗ | 0.971 | 0.934 | 0.914 |
| AMP | ✗ | 0.234 | 0.179 | 0.108 |
| Pulse | ✗ | 0.975 | 0.942 | 0.925 |
| Ours | ✓ | **0.982** | **0.977** | **0.971** |

Table 2: Interaction tasks success rate evaluation.

achieve higher success rates, their animations exhibit unnatural movements. PedAnimator's body-masked AMP and hierarchical control enable both high task success and human-like motion quality through integrated physical constraints and style preservation. More results and videos can be found in S2.6 including ablation and more visualizations or in the supplementary video.

### 4.3.2 VEHANIMATOR

As shown in 3, we evaluate the error between VehAnimator-generated animations and reference trajectories at different initial velocities to measure animation accuracy, comparing with Pure Pursuit Craig Coulter (1992)

| Methods/Speed | 0 | 5 | 10 | 20 |
|---|---|---|---|---|
| PP | 0.129/0.103 | 0.143/0.125 | 0.162/0.142 | 0.231/0.208 |
| Xu et al. | 0.075/0.054 | 0.084/0.066 | 0.095/0.077 | 0.138/0.114 |
| Ours | **0.059/0.037** | **0.062/0.041** | **0.077/0.054** | **0.106/0.088** |

Table 3: Comparison of position/velocity error.

and Xu et al. Xu & Yu (2023). VehAnimator consistently outperforms other methods in all cases. Additional experiments including ablation studies and more visualizations can be found in S2.

### 4.3.3 MULTI-LLM-AGENTS PLANNING

We benchmark LLM-agent trajectory planning against language-based traffic generation methods (LCTGen Tan et al. (2023), ChatSim Wei et al. (2024)). Due to pedestrian scenario incompatibility in other methods, evaluation focuses on vehicle-

| Methods | Language command category | | | Within road | User preference |
|---|---|---|---|---|---|
| | single | interaction | compound | | |
| LCTGen | 91.9 | 20.7 | 64.1 | 59.9 | 15.4 |
| ChatSim | 84.6 | 5.79 | 77.1 | 86.1 | 5.51 |
| Ours | **93.7** | **87.6** | **88.4** | **92.6** | **79.1** |

Table 4: High-level planning evaluation for command matching rate, within road rate, and user preferences.

only commands. Three instruction categories are evaluated: single vehicle, interaction, and composite. Using 5 maps with 26 samples per category, 1000 users assess description matching, road boundary compliance, and preference, as shown in Table 4. ChatAni achieves superior command-aligned planning and user preference. Visual comparisons in Fig. 8 (including pedestrian-involved

cases) demonstrate ChatAni's precise fulfillment of complex requirements, contrasting with LCT-Gen's limitations in interaction control and complex command execution.

### 4.4 APPLICATIONS TO AUTONOMOUS DRIVING

#### 4.4.1 AUGMENTATION FOR PREDICTION TASK

We apply ChatAni for data augmentation in two prediction tasks. For traffic prediction (minADE/minFDE), using MTR Shi et al. (2022) on the Waymo Open Dataset Sun et al. (2020), we train on 243k scenes and augment with 20% (48k) via LCTGen Tan et al. (2023) and ChatAni. Both improve minADE/minFDE, but ChatAni's superior diversity and reasonableness yielded greater gains. For human motion prediction (ADE/FDE), using HumanMAC Chen et al. (2023) on

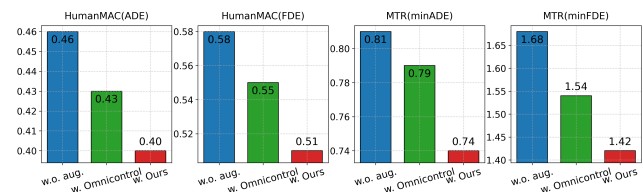

Figure 9: Prediction augmentation for traffic prediction and human keypoints prediction. All prediction tasks benefit from the enhancement provided by ChatAni, and the enhancement effect outperforms those of other baselines.

Human3.6M Ionescu et al. (2013), we augment 20% (2k) with Omnicontrol Xie et al. (2023) and PedAnimator. Both methods improve performance, but Omnicontrol's lower-quality generations constrained gains, while PedAnimator's realistic, physics-compliant outputs produced substantially better results. We use the default parameters and training settings for both methods.

#### 4.4.2 AUGMENTATION FOR VLM-BASED DRIVING SCENE UNDERSTANDING

To evaluate the impact of hazardous scenario generation on driving situations, we selected 30 scenes from the Waymo Open Dataset Sun et al. (2020) and used DriveLM Sima et al. (2024) to test collision rates 10 times per scene under three conditions: (1) original scenes, (2) ChatAni-edited scenes with intentionally heightened danger, and (3) edited scenes after fine-tuning with

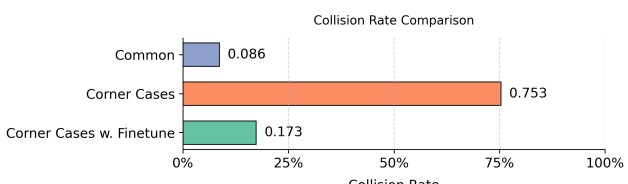

Figure 10: Collision rates of DriveLM Sima et al. (2024) under different scenarios. The data augmentation provided by ChatAni can significantly enhance the safety of VLM.

3000-frame ChatAni-augmented hazardous data. The ChatAni-edited scenes are deliberately modified to create additional hazards, which significantly increase testing collision rates. However, after augmentation and fine-tuning, the model learns to proactively decelerate or stop in these scenarios, thereby reducing collision rates. We choose the default parameters and training setting of DriveLM. See case in section S5 and supplementary video.

## 5 CONCLUSION

We propose ChatAni, the first system to achieve interactive and realistic language-driven multi-actor animation generation in street scenes. We introduce PedAnimator, a unified control policy for realistic pedestrian animation generation across multiple tasks, enabling fine-grained interactions. PedAnimator uses body-masked AMP to simultaneously improve the realism of action generation and efficiently achieve control, while employing task masking to implement a unified approach to various control methods. We introduce VehAnimator, a kinematics-based vehicle control policy with history-aware design to generate realistic vehicle animation. VehAnimator uses a learning-based approach to optimize the realism of vehicle animations based on a bicycle model, significantly enhancing robustness. ChatAni also utilizes multi-LLM-agent role-playing to enable interaction-aware high-level planning under language command. In the future, we plan to introduce more diverse traffic participants, such as cyclists.

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

## S1 ETHIC, REPRODUCILIBITY STATEMENT AND USAGE OF LLM

All related human participants involved in the user study provided consent, and there are no additional ethical concerns.

Regarding reproducibility, the experimental setup is described in the main text or appendix, with some relevant code and data included in the supplementary materials. Full code and data will be open-sourced in the future.

The LLM in this work is used for writing enhancement purposes.

## S2 DETAILS OF PEDANIMATOR

### S2.1 PEDANIMATOR STATE DETAILS

The components of the humanoid's proprioception $S_{prop}^{p}$ are as follows: joint positions $\mathbf{j} \in \mathbb{R}^{24 \times 3}$, rotations $\mathbf{q} \in \mathbb{R}^{24 \times 6}$, linear velocities $\mathbf{v} \in \mathbb{R}^{24 \times 3}$, and angular velocities $\omega \in \mathbb{R}^{24 \times 3}$ Wang et al. (2024). These components are normalized with respect to the agent's heading and root position in our simulator. The rotation $\mathbf{q}$ is represented using a 6-degree-of-freedom rotation representation. $S_{prop}^{p}$ along with the trajectory state $S_{traj}^{p}$, the motion state $S_{mo}^{p}$ to be mimicked, and the target state $S_{tar}^{p}$ for interaction, collectively forms the complete observation. During the task masking process, the remaining relevant states are masked by multiplying them by 0 based on the task to be executed, and in the motion state, specific joints can also be masked by multiplying them by 0 if necessary. For example, in the experiment, only the upper-body related states were retained.

### S2.2 TASK-RELATED REWARD AND TRAINING DETAILS

**Reward designs.** For trajectory following tasks Rempe et al. (2023), the reward is defined as $R_{\text{trajectory}} = e^{-2\|\hat{p}_t - p_t\|}$, where $\hat{p}_t$ is the position to followed at $t$, $p_t$ is the current character position. For motion imitation tasks Luo et al. (2023), $R_{\text{imitation}} = e^{-100\|\hat{j}_{pos}^t - j_{pos}^t\| \odot m^t} + e^{-10\|\hat{j}_{rot}^t - j_{rot}^t\| \odot m^t} + e^{-0.1\|\hat{j}_{vel}^t - j_{vel}^t\| \odot m^t} + e^{-0.1\|\hat{j}_{\omega}^t - j_{\omega}^t\| \odot m^t}$, where $\hat{}$ indicates the motion states to be imitated, $m^t$ is the mask to select the joints for imitation, which are the joints of upper body in our experiments. Different interaction tasks require distinct reward designs Peng et al. (2022); here, we present three sample tasks: for pushing, $R_{\text{pushing}} = 1 - u^{up} \cdot u$, $u^{up}$ is the global up vector $\mathbf{u}^*$ is the target's up vector; for patting, $R_{\text{patting}} = e^{-\|p^{rh} - c\|}$, $p^{rh}$ is the position of right hand and $c$ is the target contact position; and for walking while bending the shoulder, $R_{\text{walking\_bending}} = e^{-2\|\hat{p}_t - p_t\|} + e^{-\|p^{rh} - c\|}$, which combines the trajectory following and target position contacting. The final reward is calculated as $R = 0.5 \cdot R_{\text{disc}} + 0.5 \cdot R_{\text{task}}$.

**Interaction process.** The specific execution process for the three interaction tasks can be understood as follows: i) pushing, where the goal is to push the interaction object over, ii) patting, where the task is to gently tap a specific part of the interaction object (e.g., the shoulder) without knocking it over, and iii) walking with arm around another's shoulder, where the agent walks along a specific path while keeping the arm in contact with a specific location on the interaction object (e.g., the shoulder). During the training of the interaction tasks, the interaction object is replaced with a box in the physical environment to facilitate more stable training. However, during testing, the interaction object is replaced with another character in the simulation environment, and physical collisions and interactions are present, leading to the final output in the test phase.

**Failure recovery and crushing.** To enhance the robustness of PedAnimator, specifically to ensure that it can handle external disturbances without failing due to perturbations caused by physical collisions, we incorporate recovery during its training Luo et al. (2023). In this approach, the human pose is initialized in a collapsed or otherwise unstable standing state. Training is then conducted from this initial state, allowing the policy to learn how to recover from failure. This enables the policy to exhibit robustness when responding to physical collisions and interactions. Without this, the policy might fail to continue the action after even minor disturbances.

The crushing process is implemented for pedestrians in IsaacGym by a vehicle-like shape box. The pedestrian is crushed by the box and related animation can be readout. For the crushing vehicle, the velocity changing due to crushing is already achieved by the planning process.

### S2.3 NETWORK ARCHITECTURE

All relevant models, including the baselines, adopt the same network architecture, using an MLP as the policy network with hidden layers of 2048 and 1024 units. The final output is directed to either the latent space or the PD control signal, depending on whether hierarchical control is employed. The remaining network components, such as the discriminator, value network, control frequency (30Hz), and hyperparameters used for training, are consistent with those adopted in Pacer+ Wang et al. (2024). All training and testing are conducted on an NVIDIA 4090. The entire training process requires approximately 20 hours to fully converge.

### S2.4 EVALUATION DETAILS

**Following and imitation.** For the following and imitation tasks, we adopt the same computation methods as those used in Pacer+ Wang et al. (2024). The calculations of FID and diversity are performed using the same manual feature extraction approach as in Pacer+, with 1000 segments selected from the AMASS dataset for FID computation. For the low-speed l-FID and l-diversity, we also follow Pacer+ by testing on instances where the speed is less than 1 m/s.

**Interaction tasks.** For the three interaction-related tasks: i) pushing, the success criterion is whether the object is pushed over within the specified timestep (with a tilt along the z-axis greater than 30°), and no part of the body other than the hands is in contact with the target; ii) patting, the success criterion is whether, within the specified timestep, the right hand is within 5 cm of the target's specific location and remains there for at least 50 timesteps, with no other part of the body in contact with the target except the hands; iii) walking with arm around another's shoulder, the success criterion is that the maximum deviation from the reference trajectory is no greater than 10 cm, and the right hand is within 5 cm of the target's specific location for at least 150 timesteps. In the interaction ablation study, for the user study, we recruit 15 participants to test 30 dynamic sequences and select the ones they considered to have the highest quality.

### S2.5 RENDERING DETAILS

We utilize the rendering pipeline from ChatSim Wei et al. (2024), employing Blender's Cycles as the rendering engine. Background rendering and HDRI lighting from ChatSim are applied to the scene. The human model is based on SMPL Loper et al. (2023), with the corresponding mesh initialized, and rendered using skin and clothing textures provided in the Bedlam dataset Black et al. (2023).

### S2.6 SUPPLYMENTARY EXPERIMENTS

We provide more comprehensive visualization results to compare different aspects of PedAnimator's characteristics. All comparisons, along with animation results, can also be observed in the supplementary video material.

**Ablation study** As shown in Table S1, we also conduct ablation studies on interaction tasks to verify the effectiveness of task embedding (TE), hierarchical control (HC) and body masked AMP (M. AMP). The results indicate that, without task embedding, PedAnimator struggles to accurately identify the required interaction task, leading to difficulty in task completion. When body-masked AMP is removed, similar to AMP, the discrimination reward lacks a task completion reference. In this situation, the results can be merely close to the reference data clip without the ability to successfully execute the intended task. Without hierarchical control, the absence of action priors slows the training process and reduces the success rates.

**Imitation and following comparison.** As shown in S1, we compared the visual effects of Pacer+ during the imitation and following processes. It is evident that, under the influence of hierarchical control, PedAnimator enables smoother and more seamless transitions between different tasks—transitions that cannot be achieved with the priors provided by AMP.

Imitation and following comparison

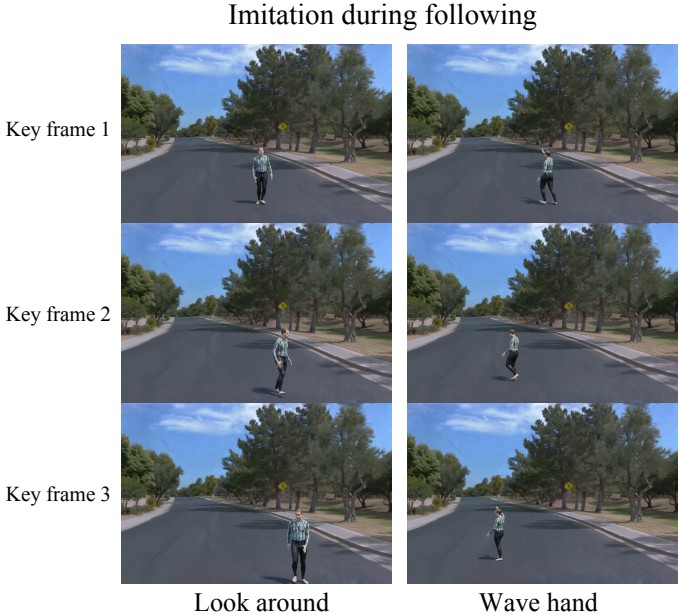

Figure S1: Comparison of imitation and following.

Imitation during following

Figure S2: Imitation during following.

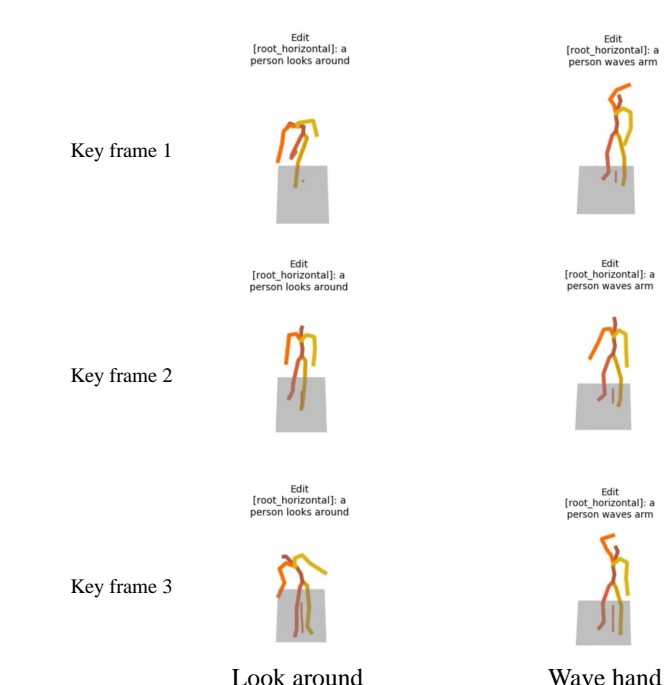

Figure S3: Failure of PriorMDM Shafir et al. (2023) with action specification during following.

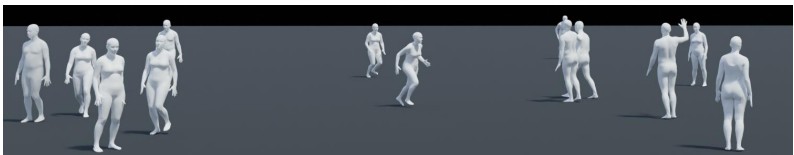

Figure S4: Animations generated by PedAnimator with MoCap, T2M, and interaction target control. PedAnimator generates diverse and realistic animations under various control signals.

**Results of imitation during following.** As shown in S2, we also provide the results of PedAnimator performing both imitation and following simultaneously. In the case of following a specific trajectory, the upper body performs actions such as looking around and waving, showcasing capability of PedAnimator to handle both imitation and following tasks concurrently.

**Interaction task comparison.** As shown in S6 S7 S8, we compared the performance of AMP, PPO, PULSE, and our method across three interaction tasks. It is still evident that while PPO and PULSE are able to complete the tasks under the given conditions, they fail to produce natural and human-like results. AMP, on the other hand, can only approximate the reference in the discriminator (walking or running), and is completely unable to complete the tasks. In contrast, our PedAnimator successfully completes the tasks while generating natural and human-like results.

**Multiple pedestrians visualization.** Fig.S4 shows PedAnimator's generation capabilities under various control modalities, including MoCap, Text2Motion-driven upper body control, trajectory control, and interactive behavior control. The system synthesizes diverse and realistic animations through planned control schemes.

### S2.7 FURTHER DISCUSSION OF KINEMATICS METHODS

For kinematics-based methods, some approaches can achieve following and motion specification, but they are completely unable to handle interaction-related tasks. Additionally, for following and motion specification tasks, these methods often suffer from overfitting to the dataset, leading to suboptimal performance. As shown in S3, the results in noticeable sliding steps and unnatural movements.

| TE | HC | M. AMP | Success rate | User preference |
|---|---|---|---|---|
| × | ✓ | ✓ | 0.403 | 0.120 |
| ✓ | ✓ | × | 0.186 | 0.069 |
| ✓ | × | ✓ | 0.948 | 0.344 |
| ✓ | ✓ | ✓ | **0.977** | **0.467** |

Table S1: Ablation study of PedAnimator. Each design contributes to the final results consistently, and the complete PedAnimator demonstrates the best performance.

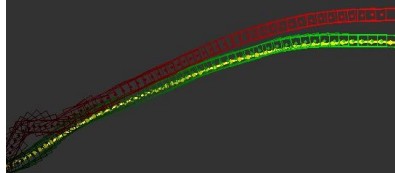 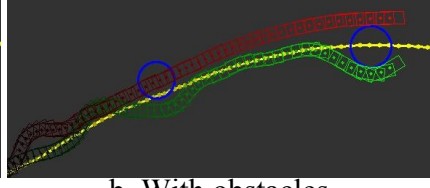

a. Without obstacles          b. With obstacles

Figure S5: Comparison of vehicle animation generation. Green boxes are our results and red boxes are Xu et al.'s Xu & Yu (2023). Yellow lines are the planned trajectory and blue circles are obstacles. VehAnimator shows better tracking results and avoid the obstacles.

## S3 DETAILS OF VEHANIMATOR

### S3.1 NETWORK ARCHITECTURE, PARAMETERS OF BICYCLE-MODEL AND TRAINING DATAILS

All networks in VehAnimator are implemented as MLPs. The policy network consists of layers with dimensions 256, 256, 128, 128, 64, and 64, while the value network has layers with dimensions 1024, 512, 256, and 128. During training, the parameters of the bicycle model (L, W, $l_f$, $l_r$) are set to two configurations: (2.7, 1.8, 0.9, 0.9) and (6.1, 2.5, 2.3, 2.0), mixed to accommodate vehicles of varying sizes. These parameters can be adjusted as needed based on specific requirements, with the two configurations provided here serving as examples. The training environment for VehAnimator was implemented through our custom codebase, with the full training process requiring approximately 10 hours on a single NVIDIA RTX 4090 GPU.

### S3.2 OBSTACLE STATE

The states of obstacles are composed of their orientation and distance relative to the vehicle's own coordinate system (considering the radius of the obstacles). The state vector is initialized with a maximum number of observable obstacles. The vector is then populated with the specific identifiers of the actual obstacles, and any remaining entries are masked. Obstacles that are too far away are directly excluded, meaning their states are not considered, and they do not contribute to the reward calculation. The distance threshold for exclusion is set to 10 in the experiment.

### S3.3 SUPPLYMENTARY EXPERIMENTS

**Ablation study.** We also conduct ablation study in Table S2, examining the effects of action filtering (Filt.), the composition of the action space (Action spa.) which means it consists of the variations or absolute scalar of velocity and steering or their variations, and the inclusion of historical states (His. state). The ablation results validate the effectiveness of each design.

**Visualization.** Furthermore, visual experiments in Figure S5 demonstrate that our method achieves more precise tracking and obstacle avoidance capabilities in scenarios with obstacles.

**Robustness.** We also provide the results using LQR Li & Todorov (2004) in S3. It can be observed that LQR is capable of vehicle animation generation from a planned trajectory to some extent. However, the planned trajectory lacks relevant constraints, which may lead to unreasonable turns or abrupt changes, causing LQR to often produce less than ideal results. Furthermore, we tested the robustness of vehicle animation generation by adding noise with a mean of 0 and variance of $\sigma$ to

| Filt. | Action spa. | His. state | Position error | Velocity error |
|:-----:|:-----------:|:----------:|:-------------:|:--------------:|
| × | × | × | 0.138 | 0.114 |
| ✓ | × | × | 0.132 | 0.111 |
| ✓ | ✓ | × | 0.115 | 0.092 |
| ✓ | ✓ | ✓ | **0.106** | **0.088** |

Table S2: VehAnimator ablation study. Each component contributes to the final result for better performance.

| Methods/Speed | 0 | 5 | 10 | 20 |
|---|---|---|---|---|
| LQR Li & Todorov (2004) | 0.074/0.058 | 0.086/0.070 | 0.092/0.079 | 0.125/0.0108 |

Table S3: Position/velocity error of LQR.

the planned trajectory. As shown in S4, the results show that LQR is highly sensitive to noise, often producing significantly worse outcomes under its influence, while other methods are relatively less affected by the noise.

**Visualization for effectiveness.** In the supplementary video, we provide results comparing the planned trajectories without using VehAnimator and with VehAnimator. It is evident that, without the involvement of VehAnimator, the animation generated by simply calculating heading between consecutive frames of the directly planned trajectory are highly unnatural, exhibiting noticeable tail swings and abrupt changes. In contrast, the results using VehAnimator are much more realistic and natural. This demonstrates the necessity of VehAnimator, as trajectories without physical constraints are highly unnatural and impractical.

## S4 DETAILS OF HIGH-LEVEL PLANNING

### S4.1 LLM-AGENT DETAILS

We provide the relevant sample prompts for the LLM agent in S10 and S11. All outputs from the LLM are in JSON format, and corresponding follow-up functions are used to convert the JSON outputs into the required data structures.

Similar to the rendering process, we also use the Waymo Open Dataset Sun et al. (2020) as the planning dataset in all experiments, and the final results are presented based on this dataset.

### S4.2 DIFFERENT LLMS

We further validated the impact of different LLMs on the results in S5. We conducted experiments using GPT-3.5 Ouyang et al. (2022) and Llama-3 Dubey et al. (2024) 70B (smaller models struggle to accurately execute the instructions). All other experimental settings remained consistent with those in the main text. It is evident that while other LLMs can handle the task to some extent, GPT-4 Achiam et al. (2023) demonstrates the most accurate understanding and decomposition of the instructions.

### S4.3 COLLISION HANDLING

In the high-level planning process, we also designed a collision handling mechanism to avoid unintended collisions. Specifically, during the trajectory generation, collision detection is performed, and when a collision is detected, a velocity adjustment function is applied to modify the speed of one of the agents. This velocity adjustment function uses a nonlinear mapping to combine the original planned result with an interpolated trajectory, leading to a planning result with different speeds. We evaluated the probability of collisions across 50 generated samples, which is calculated by dividing the number of vehicles that experienced a collision by the total number of vehicles. As shown in S6, the collision rate for all methods remains low, and ChatAni also achieves a low collision rate while incorporating collision handling.

Pushing

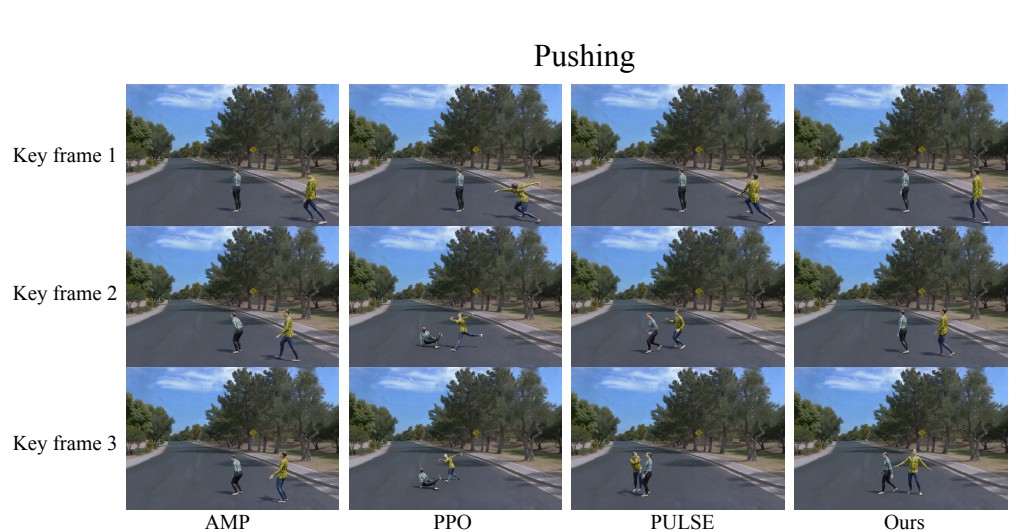

AMP          PPO          PULSE          Ours

Figure S6: Comparison of pushing.

Patting

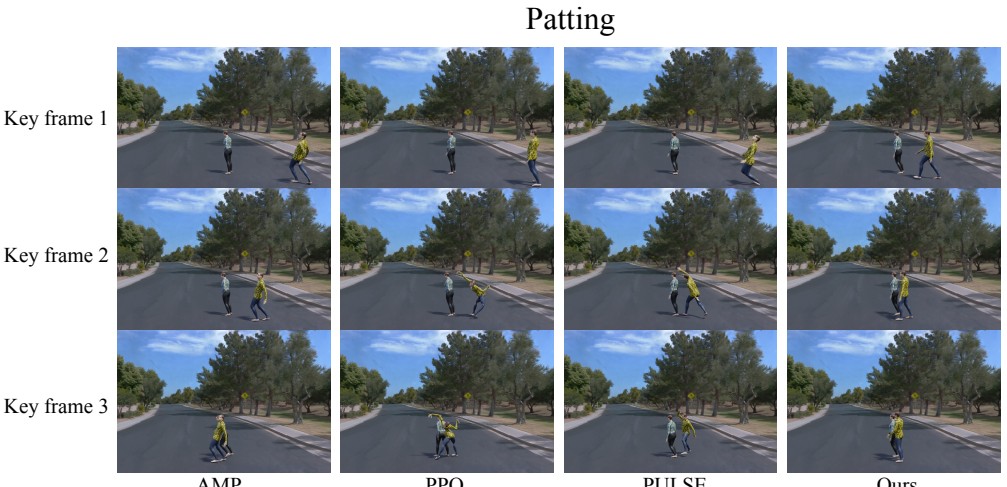

AMP          PPO          PULSE          Ours

Figure S7: Comparison of patting.

| $\sigma$ | PP Craig Coulter (1992) | Xu et al. Xu & Yu (2023) | LQR Li & Todorov (2004) | ours |
|---|---|---|---|---|
| 0.00 | 0.162/0.142 | 0.095/0.077 | 0.092/0.079 | **0.077/0.054** |
| 0.01 | 0.168/0.147 | 0.098/0.080 | 0.322/0.289 | **0.079/0.058** |
| 0.03 | 0.169/0.151 | 0.098/0.079 | 0.568/0.479 | **0.082/0.060** |

Table S4: Vehicle animation generation under Gaussian noise. $\sigma$ indicates the variance of noise.

| Methods | Language command category | | | Within road |
|---|---|---|---|---|
| | single | interaction | compound | |
| Ours-Llama3 | 0.885 | 0.742 | 0.812 | 0.920 |
| Ours-GPT3.5 | 0.854 | 0.738 | 0.834 | 0.915 |
| Ours-GPT4 | **0.952** | **0.883** | **0.896** | **0.935** |

Table S5: High-level planning evaluation for different LLMs.

## S5 CORNER CASE FOR VLM

As shown in S9, we demonstrate a corner case for Visual-Language Model (VLM)-related experiments, with the command: "Add a stationary car and have a pedestrian walk out from behind it." In this scenario, the ego vehicle initially fails to detect the pedestrian while moving forward. However, as it approaches the stationary car, it comes dangerously close to the pedestrian, leading to a collision—a classic visibility-induced corner case. Without fine-tuning, collisions occur frequently due to the system's inability to anticipate the pedestrian's presence. After fine-tuning, the vehicle proactively decelerates when significant blind zones are detected in the field of view, reducing collision probability and achieving data augmentation. The related video can be found in the supplementary video.

## S6 USER STUDY DETAILS

We conducted user studies via questionnaire format, distributing surveys containing samples from different experimental sources for participants to select the option that best meets requirements or visual preferences. All our studies involve choosing one option from multiple samples without direct rating tasks. Specific implementation details are described in their respective experiment sections, with an interface example provided in Figure S12.

## S7 LIMITATIONS

The current framework does not account for complex multi-actor animations involving cyclists, which will be addressed in future work. Additionally, fine-grained pedestrian-vehicle interactions (e.g., collision scenarios) involving intricate physical processes are reserved for subsequent research.

## S8 BORDER IMPACTS

This work serves dual purposes: (1) as a street-scene animation generator, it enhances simulation platforms with diverse, realistic pedestrian/vehicle animations for corner-case scenarios while supporting game/film production through efficient animation synthesis; (2) as a generative prior, ChatAni-produced animations can guide video generation models for controlled outputs. Technically, ChatAni pioneers the integration of multi-LLM-agent role-playing in traffic scenarios and addresses multi-pedestrian physical interactions in street environments. Key components—interaction training protocols, task-masking mechanisms, and body-masked Adversarial Motion Priors—are transferable to physics-based human animation and robotic control systems. This work demonstrates no evident potential societal negative impacts or unethical misuse scenarios.

|  | ChatSim Wei et al. (2024) | LCTGen Tan et al. (2023) | Ours |
|---|---|---|---|
| Collision rate | 0.149 | 0.092 | 0.067 |

Table S6: Collision rate of high-level planning

### Walking with arm around another's shoulder

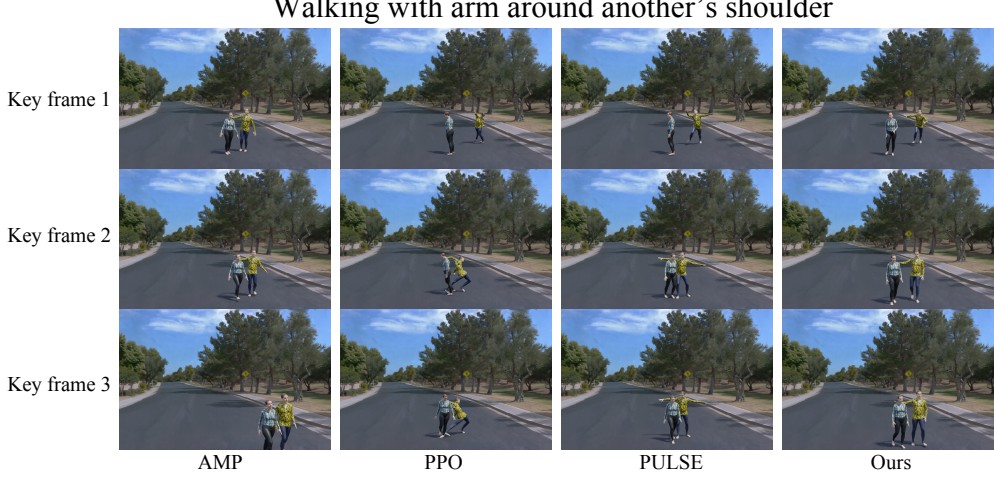

Figure S8: Comparison of walking with arm around another one's shoulder.

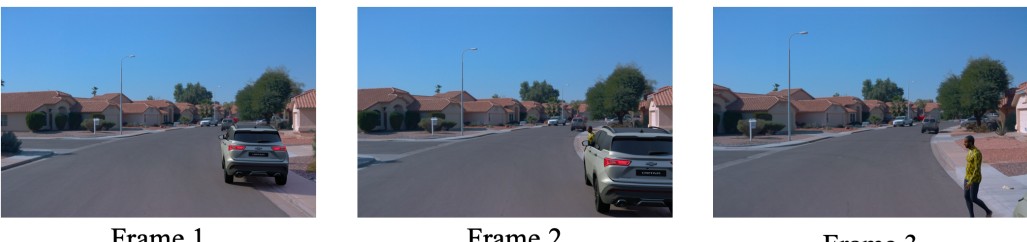

Figure S9: Corner case sample.

1242
1243
1244
1245
1246
1247
1248
1249
1250
1251
1252
1253
1254
1255
1256
1257
1258
1259
1260
1261
1262
1263
1264
1265
1266
1267
1268
1269
1270
1271
1272
1273
1274
1275
1276
1277
1278
1279
1280
1281
1282
1283
1284
1285
1286
1287
1288
1289
1290
1291
1292
1293
1294
1295

Oracle agent prompt.

I have a requirement for analyzing a scenario. I will provide you with a requirement, and I need your help to break it down into four pieces of information: (1) identify all the agents, (2) initialize each agent's state, (3) formulate a text for each agent. You should provide the information in JSON format. Note that your output should only include the JSON format of the information, not the analysis process.

(1) Identify all the agents: This means you need to extract all the agents from the scenario based on the input text. The text may be quite lengthy, so you can locate the agents by identifying the nouns, which may assist you in this task. The agents must be the objects involved in autonomous driving. The format should be as follows: agent_list = {'0': 'agent_name_0', '1': 'agent_name_1', ...}. The keys (e.g., '0', '1', etc.) represent unique agent IDs, which you should assign starting from '0'. The agents should only include vehicles like cars, pedestrians, trucks, and buses, and should not include static objects like trees or buildings. Ensure that each agent is given a distinct name. For example, in the sentence 'car a wants to overtake car b', the nouns are 'car a' and 'car b'. Both are objects in autonomous driving scenarios. Therefore, the agent_list should be formatted as follows: agent_list = {'0': 'car_a', '1': 'car_b'}. Pedestrians can be represented with identifiers like 'ped_0' and may share numbering with entities such as 'car_0'.

(2) Initialize each agent's state. In this task, you need to determine four aspects for each agent: 1. Agent type, 2. Movement, 3. Speed. You can use the results from task (1) to complete this task. Provide the initial state of each agent in a list, formatted in JSON. Most time the speed is bigger than 0. For example, the initial states should be defined as follows: init-states = ['agent_id': '0', 'agent_type': 'car', 'movement': 'overtaking', 'speed': 60, 'agent_id': '1', 'agent_type': 'car', 'movement': 'straight', 'speed': 30]. If one car intends to overtake another, it should ideally be at least twice as fast as the car it is overtaking. The initial state must include the agent type, movement, and speed.

Agent type should in [pedestrian,vehicle], action should in ['static', 'straight','pull over', 'turn over','overtake','turn left','turn right','straight left','straight right'] if it's a vehicle, and in ['static','crossing','straight'] if it is a pedestrian.

(3) Formulate a text for each agent. As an omniscient observer, you should instruct each agent on their actions through a text. The text should contain two pieces of information: 1. The agent type, 2. The agent's intention. You need to provide this in a JSON format. guide_texts = '0': 'text1', '1': 'text2', '2': 'text3', ... where the key is the agent's ID and the value is the text. The agent's name should be consistent with those in the agent_list.

Your answer should be in a JSON format,and must include the three information:agent_list,init-state,guide_texts.And init-state must include the agent type,movement and speed.

Figure S10: Oracle agent prompt.

1296
1297
1298
1299
1300
1301
1302
1303
1304
1305
1306
1307
1308
1309
1310
1311
1312
1313
1314
1315
1316
1317
1318
1319
1320
1321
1322
1323
1324
1325
1326
1327
1328
1329
1330
1331
1332
1333
1334
1335
1336
1337
1338
1339
1340
1341
1342
1343
1344
1345
1346
1347
1348
1349

### Actor agent prompt.

Now,you are an agent in the autonomous driving scenario.I will give you a text,and agent_list, describing who you are and what you need to do. Note that your output should not include your analysis process, only the JSON format of the information you provide.
I need you to analyze the text and give me the four information of the ego agent to describe what type of lane the agent should be in: (1)depend (2) speed change (3) keypoints list (4) behavior. Note that you only need to give the information of the ego agent,not the other agents.
(1) depend. You need to determine the depend of the agent according to the agent's intention. And give it like [depend_agent_id,depend_type].You can get the depend_agent_id from the agent_list. Depend_type should in ['end','start','trajectory','None'].'end' represents the ego agent's end point is the depend agent's end point, 'start' represents the ego agent's start point is the depend agent's start point, 'trajectory' represents the ego agent's whole trajectory is the depend agent's start point. If it has no depend, you should give it [-1,'None'].
(2) speed change. You need to determine the speed change of the agent according to the agent's intention. If you want to speed up, the speed change should be 1. If you want to slow down, the speed change should be -1. If you want to keep the speed, the speed change should be 0. For example, If there is someone nearby, you should slow down.
(3) keypoints list. You need to determine the number of keypoints required for your current behavior, the confirmation method for each keypoint, and their respective parameters, then return a list containing this information. Each keypoint can be confirmed in one of three ways: (1) Map-based (type '0'), with parameters: lane position ('left', 'right', or 'front'), lane type ('centerline' or 'boundary'), and driving direction ('turn left', 'turn right', or 'straight'). (2) Lane-relationship-based (type '1'), with the parameter being the relationship to the previous keypoint ('opposite direction adjacent', 'same direction adjacent', 'adjacent straight', 'adjacent left turn', 'adjacent right turn', 'different type adjacent', or 'opposite boundary'). (3) Agent-based (type '2'), with parameters specifying required information type ('point' or 'trajectory'). The result should be a sequential list of dictionaries where the key is the type ('0', '1', or '2') and the value is the corresponding parameters. For example, a left-turning car might return '['0': 'position': 'front', 'lane_type': 'centerline', 'direction': 'turn left', '1': 'relationship': 'adjacent straight']', while a car in the left lane picking up 'ped_a' might return '['0': 'position': 'left', 'lane_type': 'centerline', 'direction': 'straight', '2': 'info_type': 'point']'. Ensure compliance with parameters, common sense, and traffic regulations, with the first keypoint typically using type '0'.
(4) behavior. You need to determine your behavior. If you are a vehicle, the behavior is "None." If you are a pedestrian, the behavior corresponds to the description provided. There are two scenarios: if your behavior is a specific action such as calling or waving, simply return the text of that action; if the behavior involves interactive actions such as pushing, patting, or walking with an arm around another person, you must return the exact predefined descriptions for these three types of interactions without modification. For example, if you are calling, your behavior is "calling phone."; if you push ped_1, your behavior is "pushing." Your answer should be in a JSON format, and must include depend, speed change, keypoints list and behavior.

Figure S11: Actor agent prompt.

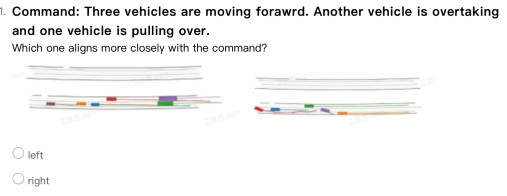

Figure S12: A sample interface of user study.

