# OpenReview forum: "ChatAni: Language-Driven Multi-Actor Animation Generation  in Street Scenes"
_ICLR.cc/2026/Conference — Submitted to ICLR 2026_

### Official Review · Reviewer_bbNe · 2025-10-28

**Soundness:** 3
**Presentation:** 3
**Contribution:** 2
**Rating:** 6
**Confidence:** 4

**Summary:**

This paper presents ChatAni, a system that generates interactive, physically plausible animations of multi-agent traffic scenes (pedestrians and vehicles) from natural-language commands. It employs a hierarchical architecture: high-level scene planning and interaction design are handled by a multi-LLM-agent framework, while low-level execution is delegated to two specialized animators—PedAnimator, a physics-based unified controller for pedestrian interactions, and VehAnimator, a kinematic controller ensuring physically valid vehicle motions. Experiments demonstrate that ChatAni can synthesize complex interactive scenarios and that its synthetic data improves downstream autonomous-driving tasks such as prediction and scene understanding.

**Strengths:**

1.	PedAnimator introduces a “body-masked AMP” strategy—adversarial motion priors with the body region occluded—that effectively reconciles the conflict between adhering to motion priors and accomplishing task-specific interactions.
2.	Rather than stopping at visually pleasing animations, the paper goes further to demonstrate the value of its synthesized data in boosting the performance of autonomous-driving prediction and scene-understanding models.
3.	The authors have constructed an end-to-end system that takes language as input and generates multi-actor animations as output.

**Weaknesses:**

1.	The so-called “language-driven interaction” in the paper is, in reality, language-based classification and invocation of a fixed set of predefined interaction types. As described (“Multi-agent interaction tasks are trained based on predefined types, with the LLM classifying interaction behaviors, enabling the animator to execute specified interactions.”), the system cannot generalize to entirely new interaction categories unseen during training. If a user were to issue an instruction such as “one pedestrian trips and another pedestrian steps forward to help,” the current framework would be unable to handle it.
2.	During interactive-task training, PedAnimator replaces the interactee with a passive “Box,” so the model effectively learns one-way object manipulation rather than genuine bidirectional interaction. This mismatch undermines the paper’s stated goal of producing physically realistic “interactions.”
3.	VehAnimator is listed as a separate contribution; however, its core merely applies the standard bicycle kinematic model to smooth trajectories and does not constitute a research-level contribution. Most of the methods in PedAnimator, apart from the body-masked AMP, likewise lack novelty.

**Questions:**

1.	Please clarify how the system would respond when confronted with a completely new interaction command that does not belong to any predefined type? Would it be necessary to redesign the reward function from scratch and retrain PedAnimator entirely? Does this imply that the system’s interaction capability is closed and non-extensible?
2.	Please clarify the theoretical or experimental basis for the claimed generalization ability of training with passive “box” objects while testing against dynamic, complex “characters”? Might this simplification in the training lead the model to learn unrealistic shortcuts that produce unnatural behavior when confronted with real characters?
3.	In the “vehicle-hit-pedestrian” scenario shown in Fig. 5, PedAnimator and VehAnimator are trained independently. How is the physical consequence of the collision transmitted and resolved between these two separate controllers? For instance, how is the vehicle’s momentum converted into an impact force on the pedestrian model? The current framework appears unable to handle such inter-agent contact dynamics in a physically consistent manner.
4.	The planning evaluation involved only 10 users, and the animation evaluation involved 15 users. Are conclusions drawn from such small sample sizes reliable? Moreover, were the participants domain experts or members of the general public? What criteria were used to assess user preferences?
5.	In the VLM scene-understanding task, the authors fine-tuned the model with only 100 frames of augmented data—an extremely small amount. The observed performance gain is more likely due to the model "memorizing" these specific hazardous scenes rather than a genuine improvement in generalization ability.

---

> ### Author Response · Authors · 2025-11-19
> **Response to questions 1-5 and weaknesses 1-3**
>
> We are grateful for your feedback and response. In response, we have organized  and merged **all the questions and weaknesses** in a structured manner.
>
> **[Q1 new interaction command : ]**
>
> In fact, we have designed a **unified** modeling of the interaction process to achieve more extensive interaction capabilities. The scenarios listed in the paper represent more **common and semantically significant** situations in road environments. The interaction process has been modeled uniformly as the observations of the **interactive targets** and the **contact positions** and **forces** between the hands and these objects. These actions can be executed according to specific directives. For instance, we provide new interaction behaviors in the **supplementary video(20s and 1min 13s)**, which do not require additional training or frameworks. Furthermore, the description regarding the modeling of the interaction process has been elaborated upon in the main text.
>
> **[Q2  training with passive “box” objects :]**
>
> We train the PedAnimator with both **initiating** and **response** of all the interaction processes. All PedAnimators are **uniform**, allowing any participating party to complete tasks using a consistent strategy. We have experimented with **directly training two characters**, but the policy used by the responding character significantly **interfered** with the training process, complicating it and leading to **instability** that made it difficult to achieve reliable results. Our **box-based** modeling and training approach successfully completed tasks during testing and proved to be easier to train, and can be **directly adopted** to two pedestrians during inference.
>
> **[Q3 How is the physical consequence of the collision transmitted and resolved : ]**
>
>
> We imported objects into the PedAnimator simulator that are nearly identical to vehicles, facilitating collision events based on the execution results of the vehicles and producing corresponding collision feedback.
>
> **[Q4 Larger user study:]**
>
> Due to limitations in time and resources during the submission process, the scale and scope of our experiments were not sufficiently extensive. Subsequently, we conduct larger-scale experiments. We **expand** the participants for both the high-level planning and PedAnimator to **100** users from the general public, testing them in the same manner as described in the main text. Users are required to assess whether the generated outcomes met the requirements of the language instructions by selecting "yes" or "no." Additionally, for preferences, users need to choose which among multiple generated results they considered to be the best. The new results indicate that our method continues to outperform other approaches, and we have updated these findings in the main text.
>
> | Methods | single | interaction | compound | Within road | User preference |
> |---------|--------|-------------|----------|-------------|----------------|
> | LCTGen  | 91.9   | 20.7        | 64.1     | 59.9        | 15.4           |
> | ChatSim | 84.6   | 5.79        | 77.1     | 86.1        | 5.51           |
> | Ours    | **93.7** | **87.6**    | **88.4** | **92.6**    | **79.1**       |
>
> | Method  | User Preference  |
> |---------|--------|
> | pacer   | 0.171  |
> | pacer+  | 0.284  |
> | ours    | **0.545** |
>
> **[Q5 Larger VLM experiments:]**
>
> Our system achieves sufficiently **precise and granular** control, allowing us to demonstrate customized enhancements tailored to very **specific scenarios and conditions**, rather than merely general improvements due to increased data volume. To validate the **general enhancements** resulting from large-scale data augmentation, we conduct additional experiments to assess the generalization improvements brought about by the increase in data volume. We select 30 scenarios for 10 repeated tests and ultimately utilized approximately 3,000 frames for augmentation. The results continue to confirm the improvements we have made to the safety of VLM.
>
> |    | Collision rate |
> |-------------|-------|
> | common      | 0.086 |
> | Corner case | 0.753 |
> | augmented   | 0.173 |
>
> **[W3 novelty of VehAnimator: ]**
>
> VehAnimator is the first method which employs a **learning-based** approach to apply a bicycle model to the VehAnimator, incorporating designs related to temporal consistency and obstacle avoidance. Experimental results, as evidenced in the **main text (Table 3) and in the appendix (Table S3)**, demonstrate that our method exhibits significantly greater **robustness and generalization** compared to the basic optimization-based bicycle model.

---

> ### Author Response · Authors · 2025-11-26
> **Summary according to different aspects**
>
> # (1) New interaction command, (2) training with boxes, (3) physical consequence of the collision, (4) larger user study, (5) larger VLM augmentation experiments, (6) novelty of VehAnimator
>
> Thank you for your valuable feedback. We have responded to **all the questions and weaknesses**, and provided a summary based on different aspects.
>
> > New interaction command (Q1)
>
> Similar to the question from other reviewers, in fact, we have designed a **unified modeling** of the interaction process to achieve more extensive interaction capabilities. The scenarios listed in the paper represent **more common and semantically significant** situations in road environments. The interaction process has been modeled uniformly as the observations of the interactive objects and the contact positions and forces between the hands and these objects. These actions can be executed according to specific directives. For instance, we provide **new interaction behaviors in the supplementary materials(20s and 1min 13s)**, which **do not require additional training or frameworks**. Furthermore, the description regarding the modeling of the interaction process has been elaborated upon in the main text.
>
> > Why training interactions with boxes? (Q2)
>
> We train the PedAnimator with both initiating and response of all the interaction processes. All PedAnimators are **uniform**, allowing any participating party to complete tasks using a consistent strategy. We have experimented with **directly training two characters**, but the policy used by the responding character **significantly interfered** with the training process, complicating it and leading to **instability** that made it difficult to achieve reliable results. Our box-based modeling and training approach **successfully completed tasks** during testing and proved to be easier to train, and can be **directly adopted** to two pedestrians during inference.
>
> > Physical consequence of the collision (Q3)
>
> We import objects into the PedAnimator simulator that are nearly identical to vehicles, facilitating collision events based on the execution results of the vehicles and producing corresponding collision feedback.
>
> > Larger user study (Q4)
>
> Due to time and resource constraints during the submission process, the scale and scope of our experiments were limited. To address this, we have since conducted **larger-scale experiments**, expanding the participant pool to 100 users from the general public for both high-level planning and PedAnimator. These experiments followed the same procedure outlined in the main text. Participants were asked to assess whether the generated outcomes met the requirements of the language instructions by selecting "yes" or "no." Additionally, they were asked to choose the best result from multiple generated options. The new results demonstrate that our method continues to outperform alternative approaches, and we have updated these findings in the main text accordingly.
> | Methods | single | interaction | compound | Within road | User preference |
> |---------|--------|-------------|----------|-------------|----------------|
> | LCTGen  | 91.9   | 20.7        | 64.1     | 59.9        | 15.4           |
> | ChatSim | 84.6   | 5.79        | 77.1     | 86.1        | 5.51           |
> | Ours    | **93.7** | **87.6**    | **88.4** | **92.6**    | **79.1**       |
>
> | Method  | User Preference  |
> |---------|--------|
> | pacer   | 0.171  |
> | pacer+  | 0.284  |
> | ours    | **0.545** |
>
> > VLM augmentation experiments (Q5)
>
> Our system offers **precise and granular control**, enabling us to demonstrate **tailored enhancements** for specific scenarios and conditions, rather than just broad improvements stemming from larger data volumes. To validate the **general benefits** of large-scale data augmentation, we conducted additional experiments to evaluate the generalization improvements achieved through increased data volume. We selected 30 scenarios, each tested 10 times, resulting in approximately 3,000 frames used for augmentation. The outcomes further confirm the improvements we have made in enhancing the safety of VLM.
> |    | Collision rate |
> |-------------|-------|
> | common      | 0.086 |
> | Corner case | 0.753 |
> | augmented   | 0.173 |
>
> > Novelty of VehAnimator (W3)
>
> VehAnimator is the first method which employs a **learning-based** approach to apply a bicycle model to the VehAnimator, incorporating designs related to temporal consistency and obstacle avoidance. Experimental results, as evidenced in the **main text (Table 3) and in the appendix (Table S3)**, demonstrate that our method exhibits significantly greater **robustness and generalization** compared to the basic optimization-based bicycle model.

---

> > ### Comment · Reviewer_bbNe · 2025-11-26
> >
> > Based on the rebuttal, i think the authors have addressed my concerns, and i will keep my positive rating.

---

> > > ### Author Response · Authors · 2025-11-26
> > >
> > > Thank you for your reply and feedback, which have helped make our work more comprehensive and complete!

---

### Official Review · Reviewer_bGam · 2025-10-31

**Soundness:** 3
**Presentation:** 3
**Contribution:** 3
**Rating:** 6
**Confidence:** 3

**Summary:**

The paper proposes a system for generating traffic scenes that are interactive (through natural language) and realistic involving both high level and low level trajectory and animation of vehicles and pedestrians. the system comprises high level LLM agent (oracle), per actor agent and low level ped, and vehicle animators.
The oracle splits the user instruction to per actor instructions and interaction groups. The actor agents then tap into the map and other actors in their interaction groups to set keypoints. The animators then realize the panned trajectory and behaviour.

**Strengths:**

(+) i see several novelties in this work: system level, i think this paper makes a stride towards a more holistic solution. work like "Language Conditioned Traffic Generation" did introduce natuaral language control but only of vehicles. This work extends it to pedestrians as well. On the other hand, works like trace and pace used physics based RL to correct for waypoint instructions but dealt only with pedestrian animation and had a policy per task. Aside for the system level improivement i identify methodological contribusion like using masking to unify policies; body masked AMP to allow naturl animation of non interacting body parts. to some extend the usage of a bicycle model to enforce plausible vehicle trajectories within a RL framework is also novel  in the context of language driven traffic control

(+) interactivity in traffic scenario generation is critical. I like that the system handles it in a layered fashion. The oracle agent establishes high level interaction groups that capture intent -- these are planned and intended interactions. Then to handle unplanned interactions that can occur during inference liek collision, the obstacle aware collision avoidance kicks in.

(+) impressive overall results, good supplementary video organization

**Weaknesses:**

(-) The system operates as a forward directed pipeline where the actor agents decide on the global trajectory before animation execution begins. while sensible, this lacks a crucial closed-loop feedback mechanism, meaning the system can only handle minor, local corrections but cannot re-generate a  new plan in response to unforeseen events,  or potential large deviations during the animation. see for example Trace and Pace where the physics based RL could feedback to the diffusion model planner as inference time guidance.

(-) i'm concerned that the reliance on the proprietary model GPT-4 API would restrics scalebility due to cost, and reproducibility. Seeing this framework inplemented with Llama would have been great

(-) The oracle agent operates purely on language inputs without explicit access to the scene's map. This risks generating high-level plans that are spatially implausible, shifting the entire burden of geometrical consistency onto the lower-level Actor Agents.

(-) Limited use cases: the authors state that "Multi-agent interaction tasks are trained based on predefined types" like pushing and patting. This constraint contradicts the claim of rich, language-driven controllability, as the system would be unable to execute or adapt to novel physical interactions outside of its fixed training categories.

**Questions:**

please refer to weaknesses

---

> ### Author Response · Authors · 2025-11-19
> **Response to weaknesses 1-4**
>
> Thank you for your valuable input and reply. We have systematically merged and responded to **all the questions and weaknesses** as per the order.
>
> **[W1 close loop simulation:]**
>
> We aim to achieve closed-loop feedback at **high-level of LLM communication** process. Within LLM communication, adjustments can be made based on the results of planning to facilitate feedback and evasive maneuvers. Furthermore, we can perform multi-step updates, allowing us to fine-tune high-level planning based on low-level generation outcomes, thereby enhancing the completeness of the closed-loop effect. However, it is worth noting that the low-level generation has already demonstrated **substantial accuracy** in executing commands; thus, we have not emphasized this aspect.
>
> **[W2 reliance on the proprietary model GPT-4 API: ]**
>
> Overall, GPT-4 exhibits stronger understanding and robustness; however, our overall design does not **rely solely** on GPT-4. The other **open-source language models** provided in Supplementary Material **Table S5** are capable of achieving favorable results and can serve as effective alternatives to GPT-4.
>
> **[W3 access to the scene's map: ]**
>
> The Oracle agent can also access map information and features, thereby assisting the actor agent in completing initial placement and planning tasks.
>
> **[W4  Limited interactions: ]**
>
> In fact, we have designed a **unified** modeling of the interaction process to achieve more extensive interaction capabilities. The scenarios listed in the paper represent more **common and semantically significant** situations in road environments. The interaction process has been modeled **uniformly** as the observations of the **interactive targets** and the **contact positions** and **forces** between the hands and these objects. These actions can be executed according to specific directives. For instance, we provide new interaction behaviors in the **supplementary video(20s and 1min 13s)**, which do not require additional training or frameworks. Furthermore, the description regarding the modeling of the interaction process has been elaborated upon in the main text.

---

> ### Author Response · Authors · 2025-11-26
> **Summary according to different aspects**
>
> # (1)  Close loop simulation, (2) reliance on GPT-4, (3) access to the scene's map, (4) limited interactions
>
> We thank you for your relevant questions. In response to **all the weaknesses** you mentioned, we have provided summarized answers and addressed them from different perspectives.
>
> > Close loop simulation (W1)
>
> We aim to achieve closed-loop feedback at **high-level of LLM communication process**. Within LLM communication, adjustments can be made based on the results of planning to facilitate feedback and evasive maneuvers. Furthermore, we can perform **multi-step updates**, allowing us to fine-tune high-level planning based on low-level generation outcomes, thereby enhancing the completeness of the closed-loop effect. However, it is worth noting that the low-level generation has already demonstrated substantial accuracy in executing commands; thus, we have not emphasized this aspect.
>
> > Reliance on the proprietary model GPT-4 API (W2)
>
> Overall, GPT-4 exhibits stronger understanding and robustness; however, our overall design does not rely solely on GPT-4. The other **open-source language models** provided in **Appendix Table S5** are capable of achieving favorable results and can serve as effective alternatives to GPT-4.
>
> > Can oracle agent access to the scene's map (W3)
>
> The Oracle agent can also access map information and features, thereby assisting the actor agent in completing initial placement and planning tasks. Some preliminary placement information for the oracle agent relies on certain data from the scene map. However, this part of the design is relatively engineering-focused and not the highlight of our work, so it is briefly omitted.
>
> > Limited interactions (W4)
>
> Similar to the question from reviewer PCmj, in fact, we have designed a **unified modeling** of the interaction process to achieve more extensive interaction capabilities. The scenarios listed in the paper represent **more common and semantically significant** situations in road environments. The interaction process has been modeled uniformly as the observations of the interactive objects and the contact positions and forces between the hands and these objects. These actions can be executed according to specific directives. For instance, we provide **new interaction behaviors in the supplementary materials(20s and 1min 13s)**, which **do not require additional training or frameworks**. Furthermore, the description regarding the modeling of the interaction process has been elaborated upon in the main text.

---

### Official Review · Reviewer_PCmj · 2025-10-31

**Soundness:** 3
**Presentation:** 3
**Contribution:** 2
**Rating:** 4
**Confidence:** 3

**Summary:**

This paper proposes ChatAni, a system designed to generate interactive and realistic traffic participant animations from natural-language instructions. The system integrates three key modules: a PedAnimator, a VehAnimator, and a Multi-LLM-Agent Role-Playing Framework. The system aims to produce controllable, multi-actor traffic animations that can serve applications such as autonomous driving simulation and behavior prediction.

The paper tackles an important and timely problem and demonstrates a well-structured and technically sound integration of several existing models. However, its novelty is unclear, many methodology descriptions are ambiguous, and results are insufficiently convincing in terms of realism, controllability, scalability, and repeatability. These concerns limit the contribution.

**Strengths:**

1. Timely and Relevant Problem:

This paper addresses the emerging challenge of language-driven multi-actor traffic animation, with clear relevance to autonomous driving and simulation research.


2. Well-Structured Framework and Nice Integration:

The paper proposes an integrated system combining pedestrian and vehicle animators with multi-LLM-agent coordination, forming a coherent overall design. It effectively merges motion generation, control policy learning, and language-based planning to enable interactive and controllable scene synthesis.

3. Practical Potential and Openness:

The paper demonstrates applicability to autonomous driving scenarios and plans to open-source code and data, enhancing the work’s potential impact.

**Weaknesses:**

1. Clarity and Technical Contributions

While the framework is well-motivated, the originality of the core techniques remains unclear. Many components appear to be adapted or directly taken from existing methods, yet the paper does not clearly indicate what has been modified or newly proposed. For example, the “Action Hierarchical Control” module sounds similar to the PULSE framework—was it directly adopted, or modified? If unchanged, this should be explicitly stated; if extended, the modifications and their impact should be clarified. Similar ambiguities appear throughout the system description, and the rationale behind some design choices is not well explained.

2. Result Quality and Script Adherence

The result videos reveal issues that raise concerns about controllability and consistency with the input scripts. In the first example, a vehicle suddenly turns sharply to avoid a pedestrian—an unscripted and abnormal event not mentioned in the textual input. The script’s instruction, “a hurried vehicle changed lane,” either does not occur or occurs at the wrong time, suggesting that the generated sequence does not faithfully follow the script. Such discrepancies indicate limited user control and weak alignment between textual instructions and generated behaviors. While the inclusion of some unscripted events can enhance realism, they should remain subtle and should not overshadow the scripted events.

3. Physical Plausibility

Despite the claim of producing physically plausible motions, several results appear unrealistic:  In the first example, a pedestrian hit by a car immediately stands up—clearly non-physical behavior. In the second example, a vehicle overtaking a taxi makes an unnecessary rightward swerve before returning to its lane, producing unnatural motion. These observations suggest that the VehAnimator’s kinematic constraints or control mechanisms are not sufficient to ensure realism in all scenarios.

4. Quantitative Accuracy and Repeatability

The paper does not discuss trajectory accuracy or repeatability. For instance: How accurately do generated trajectories follow specified positions, speeds, or timings if such quantitative details appear in the language instructions? Will the same textual script consistently produce the same animation? For autonomous driving applications—where scenario reproducibility and precision are critical—these issues significantly limit the system’s reliability and potential utility.

5. Task Scope and System Coverage

From Figure 3, it appears that PedAnimator is limited to three interaction types: pushing, patting, and arm-around-shoulder walking. If so, these assumptions should be explicitly stated and discussed. The paper demonstrates only three multi-actor scenarios, which are insufficient to validate the claimed generality. Showing a wider variety of traffic situations (e.g., crossing behaviors, group interactions, or complex multi-lane maneuvers) would strengthen the paper.

6. Evaluation and Scalability

The evaluation based on collision rate (Section 4.2.2) may not be an appropriate measure of realism. Collisions can occur due to unrealistic conditions (e.g., extreme deceleration), so a low or high collision rate alone does not indicate quality or fidelity. Moreover, the demonstrations involve fewer than ten traffic participants, and the map structures appear simplistic. Realistic street scenes often contain dozens of vehicles and pedestrians, intersections, or roundabouts. The current system’s scalability—in both scene complexity and number of participants—is unclear and should be addressed.

7. Related Work

The related work section overlooks prior simulation-based methods in multi-character and traffic animation. Claims such as
Line 107–108: “these methods do not involve interaction behaviors between multi-pedestrians or pedestrian-vehicle”
Line 116–118: “these works generally do not directly consider physical and kinematic constraints...”
are inaccurate. Several existing works explicitly consider both interaction and physical constraints, such as:

Shadow Traffic: A Unified Model for Abnormal Traffic Behavior Simulation, Computers & Graphics, 2018

Generating Believable Mixed-Traffic Animation, IEEE T-ITS, 2015

City-Scale Traffic Animation Using Statistical Learning and Metamodel-Based Optimization, SIGGRAPH, 2017

Also, the term “traffic flow” should be avoided, as it typically refers to macroscopic traffic modeling, while this paper operates at a microscopic animation level.

8. Additional Comments

Line 282: The smoothness reward combines angular and positional terms with different units; appropriate weighting seems necessary.

Line 390: The claim about “corner case simulation” lacks supporting examples.

Line 583–584: The cited paper has now been published in ICLR 2024; please update the reference.

In the third demo video, the authors should indicate which characters are controlled by which inputs (e.g., MoCAP, T2M, or interaction targets), perhaps via color labels or annotations, to improve interpretability.

**Questions:**

In the rebuttal, I would appreciate clarification on

(1) what aspects of the proposed framework are technically novel beyond system integration,

(2) how the ambiguous modules (e.g., hierarchical control) are implemented or modified from prior work,

(3) how realism and controllability are quantitatively or qualitatively evaluated,

(4) whether the system outputs are repeatable for identical input scripts, and (5) how the approach scales to more complex traffic scenes.

---

> ### Author Response · Authors · 2025-11-19
> **Response to questions 1-3 and weaknesses 1,4,6**
>
> We appreciate your comments and response. **All the questions and weaknesses** have been organized and responded to in sequence.
>
> **[Q1,W1 aspects of the proposed framework are technically novel beyond system integration : ]**
>
> Our contributions that **distinguish** this work from prior studies are summarized as follows:
>
> (1) Pedestrian Animation.
> To enhance realism, we propose a **body-masked Adversarial Motion Prior (AMP)** that improves both task completion and motion quality. A **task masking** mechanism enables unified training across different tasks. Moreover, this is among the earliest attempts to introduce **human–physics interaction** into physically based human animation.
>
> (2) Vehicle Animation.
> For vehicle animation, VehAnimator is the first to adopt a **learning-based** approach. It employs a bicycle model for trajectory post-processing to achieve smooth motion and further introduces temporal smoothing and obstacle-avoidance training mechanisms to improve dynamic consistency and realism.
>
> (3) LLM-Agent Collaboration.
> To generate diverse and semantically distinct vehicle–pedestrian animations, we design a high-level coordination mechanism based on **LLM-agent collaboration**. This enables natural language control and high-level planning for complex scene animation generation.
>
> (4) System-Level Innovation.
> From a system perspective, ChatAni is the first animation generation framework that **simultaneously models humans, vehicles, and their rich inter-agent interactions**, while supporting natural language–based scene control.
>
> Traditional methods, in contrast, typically focus only on vehicle-level planning at the high level, neglecting pedestrians and their specific behaviors, as well as the interactions among participants. At the low level, they also fail to jointly model or integrate interaction-related properties into the animation process.
>
> **[Q2 how the ambiguous modules (e.g., hierarchical control) are implemented or modified from prior work: ]**
>
> (1) The execution process of hierarchical control in our system is similar to that of PULSE. We primarily leverage hierarchical control to improve the **naturalness of transitions** between different tasks, utilizing its inherent advantages without making significant modifications to the original process.
>
> (2) Regarding the components that differ substantially from prior works:
>
> i) The **body-masked AMP** is distinct from the conventional AMP. In our design, certain body parts are intentionally ignored during AMP computation, allowing these regions to perform and complete more diverse tasks rather than merely imitating reference motions. This design improves both task accomplishment and overall motion qualitysimultaneously.
>
> ii) The **task masking** mechanism is a newly proposed approach designed to enable unified training, allowing a single model to control and execute multiple tasks efficiently.
>
> iii) For vehicle motion, VehAnimator is the first to adopt a **learning-based paradigm** which integrates a bicycle model for trajectory post-processing to achieve realistic motion. Additionally, it introduces temporal smoothing designs and a training process for obstacle avoidance, significantly improving the stability and realism of vehicle behavior.
>
> iiii) For the high-level planning component, we are among the first to explicitly consider **humans and vehicles** as different types of participants in animation. We design separate high-level planning strategies for humans and vehicles, which are then used to generate subsequent animations in a coordinated manner.
>
> **[Q3 W4 W6 how realism and controllability are quantitatively or qualitatively evaluated, Quantitative Accuracy and Repeatability, evaluation based on collision rate :]**
>
> (1) We additionally evaluated the instruction-following accuracy under specified trajectory **positions** and **velocities**. The LLM agent can accurately capture these inputs and plan trajectories that satisfy compound constraints. Although minor errors may occur during the low-level generation process, the overall execution remains reliable.
> | Method/Error  | Position | Speed |
> |---------|----------|-------|
> | LCTGen  | 8.39     | 6.33  |
> | ChatSim | 0.38     | 0.19  |
> | Ours    | **0.15** | **0.11** |
>
> (2) For the collision evaluation, we focus on **unexpected collisions** that are not mentioned in the instruction but occur during execution, as such cases represent a **violation of the given command** and indicate inadequate handling. ChatAni is capable of generating **intended collisions** when explicitly required by the instruction, and we present corresponding results to demonstrate this capability.

---

> > ### Author Response · Authors · 2025-11-19
> > **Response to questions 4-6 and weaknesses 6,2,3,5,7,8**
> >
> > **[Q4 whether the system outputs are repeatable for identical input scripts:]**
> >
> > Overall, the outputs generated under the same instruction are generally consistent. Although the LLM’s output processin high-level planning is not entirely deterministic, the results remain highly similar and closely aligned with the given instructions.
> >
> > The low-level generation process itself is **deterministic** in design, but slight variations may arise from minor numerical deviations in **physical simulation**. Even with identical inputs, these differences can occur; the system **reliably** completes the planning and adheres to the linguistic commands. These subtle variations also contribute to the diversity of the resulting animations, enriching the expressiveness and realism of the generated behaviors.
> >
> > **[Q5 W6 how the approach scales to more complex traffic scenes: ]**
> >
> > Our system can be **readily applied** to larger-scale scenarios, as the LLM possesses sufficient reasoning and planning capabilities to handle broader environments without introducing instability or performance degradation. We also provide additional experimental results, which are included in both the **main paper** and the **supplementary video(20s and 1min 13s)**, further validating the scalability and robustness of our approach.
> >
> > We aim to demonstrate **fine-grained control and detailed interactions**, enabling precise manipulation of each traffic participant’s behavior and interaction attributes. Therefore, demos in main text focus on presenting fine-grained results that showcase diverse characteristics and interaction types.
> >
> > **[W2 concerns about controllability and consistency : ]**
> >
> > The avoidance maneuver of the vehicle is also a **deliberately designed** component. We intend to ensure that collisions in the scenario occurred as **commands** rather than **accidental incidents**, aligning with user-specific requirements. The demonstration illustrates the process whereby collisions are triggered upon user demand. We consider any collision not explicitly instructed by the user as a **deviation** from the intended command.
> >
> > In the scenario, a hurried vehicle changes lanes abruptly—specifically, the blue car accelerated rapidly and executed a lane change—which corresponded **precisely** with the instructed conditions
> >
> > **[W3 physically plausible motions, VehAnimator’s kinematic constraints:]**
> >
> > The physical feasibility is reflected in the animations generated by our **physics simulation engine**, which are produced under its influence. Many physical behaviors of characters demonstrated in the demo cannot be generated through kinematics methods. The rise speed, influenced by factors such as **collision, pain, and fatigue**, is indeed a significant aspect of physical feasibility; however, it may require specific modeling to address responses in such scenarios, making it challenging to achieve solely through the physics engine. Additionally, the different motion expressions exhibited by the human body due to muscle fatigue and pain can be realized by fine-tuning certain parameters. The PedAnimator demonstrates **robustness** in responding to **falls and disturbances** while efficiently executing specific commands.
> >
> > The mentioned vehicle movements result from **high-level planning**. The Large Language Model (LLM) simulates multiple directional adjustments for vehicle parking during inference to replicate certain **real-world situations**. This is not a limitation of the VehAnimator's capabilities; rather, it reflects the LLM's ability to devise more **diverse and realistic** high-level planning possibilities.
> >
> > **[W5 Task scope and system coverage:]**
> >
> > In fact, we have designed a **unified** modeling of the interaction process to achieve more extensive interaction capabilities. The scenarios listed in the paper represent more common and semantically significant situations in road environments. The interaction process has been modeled uniformly as the observations of the **interactive objects** and the **contact positions** and **forces** between the hands and these objects. These actions can be executed according to specific directives. For instance, we provide new interaction behaviors in the supplementary materials(20s and 1min 13s), which do not require additional training or frameworks. Furthermore, the description regarding the modeling of the interaction process has been elaborated upon in the main text.
> >
> > **[W7,W8 Related work and additional comments: ]**
> >
> > Thank you for your observations. We have revised these sections in both the main text and supplementary materials, and have added relevant citations accordingly.  The smoothness reward also contains coefficients to balance the difference of units. Explanation of corner case simulation and the citation of mentioned paper are also updated.

---

> ### Author Response · Authors · 2025-11-26
> **Summary of response according to different aspects (1/3)**
>
> # (1) Technical novelties, (2) implementation of modules, (3) evaluation of realism and controllability, (4) further evaluation, (5) collision rate evaluation
>
> We sincerely thank reviewer PCmj for the comments and have provided summarized responses according to **all questions and weaknesses** from different aspects.
>
> >  Technical novelties beyond system integration(Q1, W1)
>
> Similar to the response to reviewer qBHn, in terms of **specific and reusable techniques**, the system introduces several **novel designs** distinct from prior work:
>
> - Pedestrian Animation A **body-masked AMP (Adversarial Motion Prior)** enhances task success and motion realism, while a **task masking** strategy enables unified multi-objective training. It is also among the first to integrate **human–physics interaction** into physics-based animation.
> - Vehicle Animation VehAnimator pioneers a **learning-based** framework combining a bicycle model for realistic trajectory correction with **temporal smoothing**and **obstacle-aware training**, achieving stable and natural vehicle motion.
> - Multi-Agent Planning with LLM Collaboration For diverse, behaviorally distinct vehicle–pedestrian interactions, the system employs **collaborative LLM agents** for **high-level planning** and **language-based control**, enabling flexible and semantically coherent animation generation.
> - System-Level Innovation ChatAni is the first framework to **jointly model pedestrians, vehicles, and their inter-agent dynamics**, fully controllable via natural language. Conventional methods focus mainly on vehicle planning, often neglecting pedestrians and inter-agent coordination at the animation level.
>
> > Implementation of modules (Q2, W2)
>
> (1) The execution of hierarchical control in our system follows a similar approach to PULSE. We leverage it primarily to enhance the **naturalness of transitions** between tasks, utilizing its strengths without major modifications.
>
> (2) Components that substantially differ from prior work:
>
> - **Body-masked AMP**: Unlike conventional AMP, certain body parts are ignored during computation, allowing these regions to perform diverse tasks rather than simply imitate reference motions. This improves both task success and overall motion quality.
>
> - **Task masking**: A newly proposed mechanism enabling unified training, allowing a single model to efficiently execute multiple tasks. And the training strategy and process of **physical interaction tasks** are also designed and unified by the **PedAnimator**.
>
> - **Vehicle motion (VehAnimator)**: The first learning-based framework to integrate a bicycle model for trajectory post-processing, achieving realistic motion. Temporal smoothing and obstacle-aware training further enhance vehicle stability and realism.
>
> - **High-level planning**: Explicitly distinguishes humans and vehicles as separate participants. Separate planning strategies for each are coordinated to generate coherent animations.
>
> > Evaluation of realism and controllability (Q3)
>
> We evaluated the realism using metrics including **FID** and some **user studies**, and assessed the control accuracy with the **following or tracking errors** of PedAnimator and VehAnimator, along with additional **user study** evaluations. For details, please refer to **Tables 1, 2, 3, 4**, **Figures 5, 6, 7** in the main text, as well as some **experiments and videos in the supplementary materials**.
>
> > Further evaluation of instruction-following accuracy (W4)
>
> We additionally evaluated the **instruction-following accuracy** under specified trajectory positions and velocities. The LLM agent can accurately capture these inputs and plan trajectories that satisfy compound constraints. Although minor errors may occur during the low-level generation process, the overall execution remains reliable.
> | Method/Error  | Position | Speed |
> |---------|----------|-------|
> | LCTGen  | 8.39     | 6.33  |
> | ChatSim | 0.38     | 0.19  |
> | Ours    | **0.15** | **0.11** |
>
> > Why evaluating based on collision rate (W6)
>
> For the collision evaluation, we focus on **unexpected collisions** that are not mentioned in the instruction but occur during execution, as such cases represent a **violation of the given command** and indicate **inadequate handling**. ChatAni is capable of generating **intended collisions** when explicitly required by the instruction, and we have presented corresponding results in the demo to demonstrate this capability.

---

> ### Author Response · Authors · 2025-11-26
> **Summary of response according to different aspects (2/3)**
>
> # (6) Repeatable system output, (7) crowded and complex generation, (8) Why avoidance, (9) a hurried vehicle does not occur, (10) physically plausible motions
>
> > Repeatable system output for identical input scripts (Q4)
>
> Overall, the outputs generated under the same instruction are **generally consistent**. Although the LLM’s output processin high-level planning is not entirely deterministic, the results remain highly similar and closely **aligned with the given instructions**. The low-level generation process itself is deterministic in design, but **slight variations** may arise from minor numerical deviations in **physical simulation**. Even with identical inputs, these differences can occur; nevertheless, the system reliably completes the planning and adheres to the linguistic commands. These subtle variations also contribute to the **diversity** of the resulting animations, enriching the **expressiveness and realism** of the generated behaviors.
>
> >  More experiments about crowded and complex environments (Q5, W6)
>
> Similar to the response to reviewer qBHn, additional results in the **main text and supplementary video (20s, 1min 13s)** demonstrate strong **scalability, robustness**, and the ability to interpret abstract instructions for generating complex, crowded scenes. Large-scale scenes can be **directly processed** by our system, as the Oracle agent effectively handles complex macroscopic environments.
> To highlight **fine-grained and precise control**, demos in the main text focus on refined language-based manipulation, showcasing diverse interaction types and behavior control patterns. Meanwhile, the capability to handle larger scenes of ChatAni is also sufficient.
>
> > Why avoidance? Concerns about controllability and consistency? (W2)
>
> Silimar to evaluating based on collision rate, The avoidance maneuver of the vehicle is also a **deliberately designed component**. We intend to ensure that collisions in the scenario **occurred as commands rather than accidental incidents**, aligning with user-specific requirements. The demonstration illustrates the process whereby collisions are **triggered upon user demand**. We consider any **collision not explicitly instructed by the user as a deviation** from the intended command.
>
> The avoidance function can also be easily **deactivated** if we want the Animators to execute the planning directly.
>
> > “a hurried vehicle changed lane,” does not occur? (W2)
>
> In the scenario, a **hurried vehicle changes lanes abruptly**—specifically, **the blue car** accelerated rapidly and executed a lane change—which corresponded precisely with the instructed conditions. The frames in the figures can not show the entire process but you can check the **supplementary video** to find the demo video.
>
> > physically plausible motions (W3)
>
> The physical feasibility is reflected in the animations generated by our **physics simulation engine**, which are produced under its influence. Many physical behaviors of characters demonstrated in the demo **cannot be generated through kinematics methods**. The rise speed, influenced by factors such as **collision, pain, and fatigue**, is indeed a significant aspect of physical feasibility; however, it may require specific modeling to address responses in such scenarios, making it challenging to achieve solely through the physics engine. Additionally, the different motion expressions exhibited by the human body due to muscle fatigue and pain can be realized by fine-tuning certain parameters. The PedAnimator demonstrates **robustness in responding to falls and disturbances** while efficiently executing specific commands.

---

> ### Author Response · Authors · 2025-11-26
> **Summary of response according to different aspects (3/3)**
>
> # (11) VehAnimator’s kinematic constraints, (12) task scope and system coverage, (13) smoothness reward, (14) claim about “corner case simulation”, (15) related works and citations
>
> > VehAnimator’s kinematic constraints (W3)
>
> The mentioned vehicle movements result from **high-level planning**. The Large Language Model (LLM) simulates multiple directional adjustments for vehicle parking during inference to replicate **certain real-world situations**. This is not a limitation of the VehAnimator's capabilities; rather, it reflects the LLM's ability to devise **more diverse and realistic high-level planning possibilities**.
>
> > Task scope and system coverage (W5)
>
> In fact, we have designed a **unified modeling** of the interaction process to achieve more extensive interaction capabilities. The scenarios listed in the paper represent **more common and semantically significant** situations in road environments. The interaction process has been modeled uniformly as the observations of the interactive objects and the contact positions and forces between the hands and these objects. These actions can be executed according to specific directives. For instance, we provide **new interaction behaviors in the supplementary materials(20s and 1min 13s)**, which **do not require additional training or frameworks**. Furthermore, the description regarding the modeling of the interaction process has been elaborated upon in the main text.
>
> > Smoothness reward (W8)
>
> he smoothness reward also contains **coefficients** to balance the difference of units. We polish the writing details in the main text.
>
> >  Claim about “corner case simulation” (W8)
>
> Some dangerous behaviors are demonstrated in the demo, such as pushing between pedestrians or some dangerously avoidance. These behaviors are not common but very dangerous so we call them corner cases. We revise and polish the details in the main text.
>
> > Related works and citations (W7,W8)
>
> Thank you for providing these additional materials. We have incorporated the references into the main text and made the necessary revisions. However, we also acknowledge that these works have several **limitations compared to ChatAni**, and we have addressed this in the main text as well.

---

### Official Review · Reviewer_qBHn · 2025-11-01

**Soundness:** 3
**Presentation:** 3
**Contribution:** 2
**Rating:** 4
**Confidence:** 4

**Summary:**

This paper introduces ChatAni, a system designed to generate multi-agent traffic trajectories involving both pedestrians and vehicles. ChatAni leverages a large language model (LLM) to produce high-level commands and initializations for each participant. These commands are then passed to reinforcement learning (RL)-based low-level control policies that generate detailed motion behaviors.

**Strengths:**

1. The paper addresses an interesting and relevant problem in multi-agent simulation and animation.

2. The manuscript is clearly written and easy to follow.

**Weaknesses:**

1. The technical novelty appears limited. The system seems to be a relatively straightforward integration of existing components: LLMs for high-level reasoning and AMP/RL-based controllers for low-level motion generation. Both of these components have been widely studied in the literature. The main contribution appears to be the composition of these modules into a traffic simulation framework rather than the introduction of a new algorithmic or conceptual advance.

2. The scalability of the approach is unclear. The presented experiments involve only a small number of entities, leaving open questions about whether the system can handle large-scale scenarios with dozens or hundreds of agents.

3. There are still some dirty, hard-coded components, such as the collision handling in appendix S4.3.

**Questions:**

1. Could the authors clarify what differentiates ChatAni from prior systems that combine high-level planning (via LLMs) with low-level control (via RL)? What are the key technical insights or contributions that make this work novel?

2. Have the authors attempted to scale the system to more complex and crowded environments, such as airport pickup zones or train stations, where random events like pullovers, loading/unloading, or emergent interactions occur?

3. Can ChatAni generate or handle long-tail corner cases—such as a vehicle door opening unexpectedly and obstructing others? Demonstrating such robustness or diversity would significantly strengthen the empirical results and highlight the system’s potential beyond proof-of-concept integration.

---

> ### Author Response · Authors · 2025-11-19
> **Response to questions 1-3 and weaknesses 1-3**
>
> Thank you for your feedback and response！ We have organized **all the questions and weaknesses** in order, providing our replies accordingly.
>
> **[Q1,W1 The key technical insights or contributions :]**
>
> In terms of **specific and reusable techniques**, the system introduces several **novel designs** which are different from prior works:
>
> (1) Pedestrian Animation.
> To enhance realism, the system employs a **body-masked AMP** (Adversarial Motion Prior) mechanism, which improves both task completion and motion quality. Furthermore, a \textbf{task masking} strategy enables unified training across different animation objectives. Moreover, this is among the earliest approaches to introduce **human–physics interaction** into physically-based animation frameworks.
>
> (2) Vehicle Animation.
> For vehicle motion, VehAnimator is the first to adopt a **learning-based** paradigm which integrates a bicycle model for trajectory post-processing to achieve realistic motion. Additionally, it introduces temporal smoothing designs and a training process for obstacle avoidance, significantly improving the stability and realism of vehicle behavior.
>
> (3) Multi-Agent planning with LLM Collaboration.
> To generate diverse and behaviorally distinct vehicle–pedestrian animations within complex scenes, the system incorporates a high-level mechanism driven by **LLM agents collaboration**. This framework enables natural language control and high-level planning, facilitating flexible and semantically aligned animation generation.
>
> (4) System-Level Innovation.
> From a system perspective, ChatAni represents the first animation generation framework that **simultaneously models pedestrians, vehicles, and their rich inter-agent interactions** while allowing natural language–based scene control.
> Traditional methods, by contrast, typically focus on vehicle behavior at the high-level planning stage while neglecting the role of pedestrians and their specific actions. Furthermore, they often fail to capture inter-agent interactions or integrate such properties into low-level animation control.
>
>
> **[Q2,W2 more complex and crowded environments:]**
>
> Large-scale scene processing can be **directly handled** by our system, as the Oracle agent remains capable of managing broader and more complex macroscopic environments. Additional experimental results are provided in both the **main text** and the **supplementary video(20s and 1min 13s)**, showing that our method possesses strong **scalability and robustness**, and can reason over abstract instructions to generate complex and crowded outcomes.
>
> To emphasize more **specific, precise, and fine-grained control**, demos in main text mainly demonstrate refined language control capabilities. To present diversified animations, they primarily showcase different types and characteristics of interactions and behavior control.
>
> **[Q3 vehicle door opening unexpectedly and obstructing others:]**
>
> The case of opening doors is not part of the animation design itself but rather relates to the **hinge structure** of specific 3D assets. Our system can **easily integrate** such capabilities if appropriate digital assets are provided; however, this is not among our technical contributions or highlights. The results in the supplementary video also **contain more dangerous and corner behaviors** in the street scenes, such as pushing between pedestrians, vehicle crashes pedestrian and vehicle yielding to pedestrians, which showcase the capability of ChatAni to generate long-tail corner cases.
>
> The animation generation focuses on the **dynamics** of traffic participants—such as vehicles and pedestrians—to construct their dynamic behaviors, rather than modifying the digital assets themselves. Besides, we additionally demonstrate richer interactive behaviors to show that ChatAni is capable of generating more diverse and effective animations.
>
> **[W3 some dirty, hard-coded components:]**
>
> The LLM agent is equipped with different tools to perform various operations. The obstacle avoidance module described in Appendix S4.3 is one effective tool that can handle relevant situations efficiently. However, this module is **not a relatively critical component** of the system, as the primary focus remains on fine-grained animation generation itself.
> The LLM agent is equipped with different tools, each designed to accomplish a specific function, which together form a coherent and efficient system. For components that can be implemented in a straightforward manner, we adopt **simple yet effective solutions**. Nevertheless, ChatAni still incorporates a **wide range of novel and reusable technical innovations**.

---

> ### Author Response · Authors · 2025-11-26
> **Summary of response according to different aspects**
>
> # (1) Technical insights,  (2) crowded and complex generation, (3) vehicle door opening unexpectedly and (4) hard-coded components
>
> We sincerely thank reviewer qBHn for the valuable comments. Based on the key points raised in **all the questions and weaknesses sections**, we have summarized our responses from different aspects.
>
> > The key technical insights or contributions（Q1, W1）
>
> In terms of **specific and reusable techniques**, the system introduces several **novel designs** distinct from prior work:
>
> - Pedestrian Animation A **body-masked AMP (Adversarial Motion Prior)** enhances task success and motion realism, while a **task masking** strategy enables unified multi-objective training. It is also among the first to integrate **human–physics interaction** into physics-based animation.
> - Vehicle Animation VehAnimator pioneers a **learning-based** framework combining a bicycle model for realistic trajectory correction with **temporal smoothing**and **obstacle-aware training**, achieving stable and natural vehicle motion.
> - Multi-Agent Planning with LLM Collaboration For diverse, behaviorally distinct vehicle–pedestrian interactions, the system employs **collaborative LLM agents** for **high-level planning** and **language-based control**, enabling flexible and semantically coherent animation generation.
> - System-Level Innovation ChatAni is the first framework to **jointly model pedestrians, vehicles, and their inter-agent dynamics**, fully controllable via natural language. Conventional methods focus mainly on vehicle planning, often neglecting pedestrians and inter-agent coordination at the animation level.
>
> > More experiments about crowded and complex environments(Q2, W2)
>
> Additional results in the main text and **supplementary video (20s, 1min 13s)** demonstrate strong **scalability, robustness**, and the ability to interpret abstract instructions for generating complex, crowded scenes. Large-scale scenes can be **directly processed** by our system, as the Oracle agent effectively handles complex macroscopic environments.
> To highlight **fine-grained and precise control**, demos in the main text focus on refined **language-based manipulation**, showcasing diverse interaction types and behavior control patterns. Meanwhile, the capability to handle larger scenes of ChatAni is also sufficient.
>
> > Vehicle door opening unexpectedly(Q3)
>
> The door-opening case depends on the **hinge structure** of specific 3D assets rather than our animation design. While our system can **readily incorporate** such functions with suitable assets, this lies outside our **technical focus**. The results in the supplementary video also contain more dangerous and corner behaviors in the street scenes, such as pushing between pedestrians, vehicle crashes pedestrian and vehicle yielding to pedestrians, which showcase the capability of ChatAni to generate long-tail corner cases
> Our animation generation targets the **dynamics of traffic participants**—both vehicles and pedestrians—to model their motion behaviors. Additionally, we present richer **interactive behaviors** to demonstrate ChatAni’s capability for **diverse and effective animation generation.**
>
> > Hard-coded components (W3)
>
> The **LLM agent** is equipped with a suite of specialized tools, each designed to accomplish a specific function, collectively forming a **cohesive and efficient system**. Among them, the obstacle avoidance module (Appendix S4.3) serves as an **effective component** for handling relevant scenarios. However, since this function can be implemented through relatively simple means, it is **not a core focus** of our framework.
> The LLM agent is equipped with different tools, each designed to accomplish a specific function, which together form a coherent and efficient system. For components that can be implemented in a straightforward manner, we adopt **simple yet effective solutions**. Nevertheless, ChatAni still incorporates a **wide range of novel and reusable technical innovations**.

---

### Author Response · Authors · 2025-12-03
**Summary for AC (1/3) : Strengths**

# Summary for AC (1/3) : Strengths

We would like to express our **sincere appreciation** to the AC for their dedicated efforts in this conference. A consolidated summary has been prepared based on all reviewers' feedback to assist the AC in efficiently reviewing the key points.

---

## 1: An **interesting systems design** that investigates a **novel and important problem**. (Mentioned by all reviewers)

All reviewers noted that our work, ChatAni, presents an **engaging and comprehensive system** that is the first to fully model **fine-grained animation generation** for interactions including both pedestrians and vehicles in street scenarios, with the capability of being directly controlled and driven by natural language. ChatAni **addresses the problem holistically** and delivers all components of the system and incorporates multiple novel technical designs and innovations tailored to the requirements.

---

## 2: Promising future **application value** and **impact**.（Mentioned by reviewers PCmj， bbNe，bGam）

3 reviewers highlighted, aligning with our original design intent, that ChatAni can provide **efficient and effective data generation for autonomous driving** and other application scenarios. It is capable of simulating hazardous and hard-to-collect corner cases, and its utility has been partially validated through experiments.

---

## 3: Numerous specific and reusable **technological innovations**. (Mentioned by reviewers  bbNe，bGam)

ChatAni introduces a suite of **distinct technical innovations** across its components to enhance final output quality and address specific challenges.
- For pedestrian animation, we designed a **body-masked AMP** to improve motion realism while fulfilling tasks, alongside a novel **task-masking** mechanism and training strategy enabling unified end-to-end training. Our work also pioneers the modeling of **physical interactions in multi-pedestrian** animation generation.
- For vehicle animation, we propose the first **earning-based** approach that effectively and robustly generates animations using a bicycle model, supplemented by multiple detailed optimizations to refine the final results.
- At the system level, we developed a **multi-LLM-agent collaboration framework** tailored for road scenes, facilitating holistic scenario planning and enabling flexible, precise control via natural language.
- Collectively, these innovations form a **comprehensive system** that effectively addresses practical and meaningful real-world requirements.

---

## 4. **Impressive results** (Mentioned by reviewers  bbNe，bGam)

ChatAni was also noted for its impressive final demonstrations, comprehensive comparative experiments, and validation of its application scenarios.

---

> ### Author Response · Authors · 2025-12-03
> **Summary for AC (2/3) : Common questions and our explanations**
>
> # Summary for AC (2/3) : Common questions and our explanations
>
> We have **addressed all questions and weaknesses** raised by the reviewers, providing detailed responses and further clarification. Below, we summarize the **key common themes** and provide our **consolidated explanations**. For more specific point-by-point responses, please refer to the **separate replies to each reviewer**.
>
> We also uploaded the **revised paper and supplementary material** to show more information. We have highlighted the revisions using **different text colors**.
>
> ---
>
> ## 1. **Technical novelties** and differences from priors works (Mentioned by reviewr qBHn and PCmj)
>
> In fact, the other two reviewers have both expressed **strong recognition for ChatAni's technical innovations**—either in their official reviews or during preliminary discussions—highlighting them as **key strengths**. ChatAni goes far **beyond simply integrating a complete system**; it incorporates numerous reusable technical innovations that are distinct from existing work. Together, these form a coherent and fully functional system that addresses a novel yet important problem, with significant practical potential. Below, we reiterate the specific **technical contributions of ChatAni beyond system integration**—all of which are novel compared to prior work and reusable in future research:
> - (1) Pedestrian Animation A **body-masked AMP (Adversarial Motion Prior)** enhances task success and motion realism, while a **task masking** strategy enables unified multi-objective training. It is also among the first to integrate **human–physics interaction** into physics-based animation.
> - (2) Vehicle Animation VehAnimator pioneers a **learning-based** framework combining a bicycle model for realistic trajectory correction with **temporal smoothing**and **obstacle-aware training**, achieving stable and natural vehicle motion.
> - (3) Multi-Agent Planning with LLM Collaboration For diverse, behaviorally distinct vehicle–pedestrian interactions, the system employs **collaborative LLM agents** for **high-level planning** and **language-based control**, enabling flexible and semantically coherent animation generation.
> - (4) System-Level Innovation ChatAni is the first framework to **jointly model pedestrians, vehicles, and their inter-agent dynamics**, fully controllable via natural language. Conventional methods focus mainly on vehicle planning, often neglecting pedestrians and inter-agent coordination at the animation level.
>
> ---
>
> ## 2. **Scalability** of system (Mentioned by reviewr qBHn and PCmj)
>
> The reviewers raised questions regarding ChatAni's performance in **complex and crowded scenarios**. In response, we have supplemented relevant experiments in both the **main text and supplementary materials(video at 20s)** to validate its **scalability**. The results demonstrate that ChatAni remains fully capable of handling large and complex environments while producing rich and diverse generation outcomes.
>
> Furthermore, while ChatAni is designed to support **precise control for customized animation generation**—focusing on fine-grained semantic accuracy rather than abstract macro-level semantics, as emphasized in our original experiments—it inherently exhibits strong scalability. The system can seamlessly adapt to large-scale replicated scenarios without requiring additional adjustments, as confirmed by the newly added experiments.
>
> ---
>
> ## 3. Capability of handling **novel interactive behaviors** (Mentioned by reviewr bbNe, qBHn and PCmj)
>
> In fact, we have designed a **unified modeling** of the interaction process to achieve more extensive interaction capabilities. The scenarios listed in the paper represent **more common and semantically significant** situations in road environments. The interaction process has been modeled uniformly as the observations of the interactive objects and the contact positions and forces between the hands and these objects. These actions can be executed according to specific directives. For instance, we provide **new interaction behaviors in the supplementary materials(20s and 1min 13s)**, which **do not require additional training or frameworks**. Furthermore, the description regarding the modeling of the interaction process has been elaborated upon in the main text.
>
> ---
>
> ## 4. **Additional experiments** are required to be supplemented or modified.  (Mentioned by reviewr bbNe and PCmj)
>
> All requested additional experiments and experimental validations mentioned in the reviews have been **incorporated into the main text or supplementary materials**. Corresponding responses have also been provided **in the rebuttal**. Additionally, relevant experiments that the reviewers may have overlooked have been explicitly **pointed out** in the replies to each comment.

---

### Meta-Review · Area_Chair_jFo9 · 2026-01-11

**Summary:**

This paper introduces a language-driven system for generating interactive, realistic multi-actor street-scene animations by combining a multi-LLM-agent planner with two physically-based animators (PedAnimator and VehAnimator). Reviewers agree the problem is practically relevant and the system is well-designed. Initial concerns on novelty relative to prior works, scalability to complex scenes and long-tail behaviors are reasonably addressed by the author response. But concerns remain on limited technical novelty beyond system design, lack of physical plausibility in some examples and insufficient evaluation of controllability and repeatability. Given these important remaining concerns, the paper is not recommended for acceptance to ICLR. The authors are encouraged to include all review suggestions for submission to a future venue.

**Reviewer Concerns:**

### Addressed concerns

* **qBHn:** The novelty of the system relative to prior LLM and RL animation frameworks was unclear. The author response details specific technical contributions (such as body-masked AMP, VehAnimator with bicycle-model-based learning and multi-agent LLM collaboration) and positions the work as the first to jointly model pedestrians, vehicles and their interactions under language control.

* **qBHn:** The scalability of the system to crowded and complex environments is uncertain. The author response points to additional experiments and supplementary videos demonstrating larger scenes with more participants.

* **qBHn:** The handling of long-tail or dangerous behaviors was questioned. The author response points to supplementary results with pedestrian-vehicle interactions, crashes, yielding and other hazardous scenarios.

* **bbNe:** Scale of user study and VLM experiments. The author response adds these further experiments.


### Unaddressed concerns

* **bGam:** Closed-loop simulation. The author response states this is achieved by the "LLM communication process", but this is not substantially supported by descriptions or experiments in the paper.

* **PCmj:** The technical originality remains unclear, as many components appear adapted from existing methods. While the author response lists multiple innovations, it does not fully resolve whether the core contributions are algorithmic in nature.

* **PCmj:** Several results show limited adherence to the input script (such as unscripted vehicle maneuvers, missing or mistimed commanded behaviors). The author response argues that some deviations enhance realism and that other examples in the videos satisfy the commands, but it does not establish controllability, which should be an important dimension to analyze quantitatively.

* **PCmj:** Physical plausibility is not always convincing (such as pedestrians immediately standing after collisions, unnatural vehicle maneuvers). The author response explains design choices and notes configurability, but does not resolve the efficacy of the kinematic constraints or control mechanisms.

* **PCmj:** Quantitative accuracy, repeatability and reproducibility are insufficiently evaluated. The author response does not add controlled experiments demonstrating reproducibility or quantitative adherence to specified positions, speeds, or timings.

* **qBHn:** Some system components remain hard-coded (such as obstacle handling). The author response considers these as non-core utilities, but this does not fully address robustness or generality.

**Reviewer Scores:**

* **qBHn:** Initial rating 4, concerns not fully addressed, likely to maintain at 4.
* **PCmj:** Initial rating 4, concerns not fully addressed, likely to maintain at 4.
* **bGam:** Initial rating 6, concerns are mostly addressed, likely to maintain at 6.
* **bbNe:** Initial rating 6, likely to maintain at 6.

---

### Decision · Program_Chairs · 2026-01-26

Reject